# Kernel Trace Distance: Quantum Statistical Metric between Measures through RKHS Density Operators

## Abstract

Distances between probability distributions are a key component of many statistical machine learning tasks, from two-sample testing to generative modeling, among others. We introduce a novel distance between measures that compares them through a Schatten norm of their kernel covariance operators. We show that this new distance is an integral probability metric that can be framed between a Maximum Mean Discrepancy (MMD) and a Wasserstein distance. In particular, we show that it avoids some pitfalls of MMD, by being more discriminative and robust to the choice of hyperparameters. Moreover, it benefits from some compelling properties of kernel methods, that can avoid the curse of dimensionality for their sample complexity. We provide an algorithm to compute the distance in practice by introducing an extension of kernel matrix for difference of distributions that could be of independent interest. Those advantages are illustrated by robust approximate Bayesian computation under contamination as well as particle flow simulations.

## 1 INTRODUCTION

Statistical distances are ubiquitous in the fundamental theory of machine learning and serve as the backbone of many of its applications, such as: discriminating between the generative model and real data in Generative Adversarial Networks (GAN) [Goodfellow et al., 2014, Arjovsky et al., 2017, Li et al., 2017, Genevay et al., 2018, Birrell et al., 2022], testing whether a dataset is close to another (two-sample test) [Eric et al., 2007, Gretton et al., 2012, Hagrass et al., 2024] or to a particular distribution (goodness-of-fit test), as well as acting as an objective loss function in particle gradient flows [Arbel et al., 2019, Feydy et al., 2019, Korba et al., 2021, Hertrich et al., 2023, Neumayer et al., 2024, Chen et al., 2024], or in minimum distance estimators [Wolfowitz, 1957, Basu et al., 2011].

A class of distances between probability distributions, called Integral Probability Metrics (IPM) [Müller, 1997], is defined by measuring the supremum of difference of integrals over a function space. It comprises many popular metrics such as the Total Variation distance, Wasserstein-1 distance and the Maximum Mean Discrepancy (MMD) [Gretton et al., 2012] also known as quadratic distance [Lindsay et al., 2008]. IPMs' theoretical properties were largely investigated in the literature, such as their statistical convergence rate [Sriperumbudur et al., 2010], concentration for inference using ABC [Legramanti et al., 2022], PAC-Bayes bounds [Amit et al., 2022], as well as adversarial interpretations [Husain and Knoblauch, 2022]. For instance, the MMD enjoys a fast statistical convergence rate of $O(n^{-\frac{1}{2}})$ while the Wasserstein distance suffers from the curse of dimensionality with a rate no better than $\Theta(n^{-\frac{1}{d}})$ [Kloeckner, 2012]. One could wonder: *how large such a function space could be before the curse of dimensionality kicks in?* In this work, we theoretically investigate how to get closer to such frontier by defining an extended family of kernel distances, that write as novel IPM whose dual function space is larger than the one of MMD.

Kernel methods allow to represent a distribution by a vector by associating to a datapoint $x$ a feature map image $\varphi(x)$ in a Hilbert space, and by doing so, embed in a linear way a distribution $\mu$ to what is called a *(kernel) mean embedding* $\mathbb{E}_{X \sim \mu}[\varphi(X)] = \int \varphi(x) d\mu(x)$. However mean embeddings for different distributions may have different "energies", i.e., squared Hilbert norms, which may lead to several pitfalls of MMD. In quantum information theory [Watrous, 2018], a similar idea to mean embedding is called superposition. The quantum equivalent of a datapoint or deterministic Dirac distribution is called a *pure state* and is a projector of rank and trace one, that could be denoted $vv^*$ (or $|v\rangle\langle v|$) for a unit vector $v$. Its analog for a general probability distribution is called a *mixed state* and is the superposition $\sum_v p(v)|v\rangle\langle v|$ where $p(v)$ are probabilities. A non-trivial mixed state can hardly be confused with a pure state as a linear combination of different projectors is of higher rank than 1: using projecting operators instead of the vectors themselves makes the *linearity less "trivial"*. As those positive definite operators can be diagonalised, by using always the same orthogonal basis and studying the eigenvalues, we recover classical probabilities, and as such we can see quantum probabilities as their extension. Recently, the work of Bach [2022] introduced a novel divergence between probability distributions, by plugging a kernel operator embedding of the distributions (which are also positive definite operators) in the Von Neumann relative entropy from quantum information theory (i.e., a Kullback-Leibler divergence between positive Hermitian operators), and whose statistical and geometrical properties were investigated more in depth in Chazal et al. [2024]. Instead of considering a divergence on such operators, here we propose to draw inspiration from quantum statistical metrics, which enjoy nice geometrical properties such as the triangle inequality. Two of them are well-known and mutually bounding: the Bures metric, and the trace distance, on which we focus here, and which is derived from a (Schatten) norm.

**Related works** The kernelised version of Bures metric, i.e., a Bures metric between kernel covariance operators, has been studied for instance in Oh et al. [2020], Zhang et al. [2019]. The closest work to ours is the one by Mroueh et al. [2017]. They consider a similar metric to ours, i.e. the trace distance, that they refer to as Covariance Matching IPM. It shares the same dual writing as the metric we consider, yet, in that work, the dual problem is solved through a numerical program involving neural networks that approach kernel features. Hence, they compute an approximate version of their target metric. In contrast, we use kernel features directly in the dual formulation, and derive a closed-form for the metric leveraging a kernel trick. Moreover, we provide theoretical guarantees regarding this metric and investigate different numerical applications than the one of the GAN considered in Mroueh et al. [2017].

**Contributions** Our main contributions can be summarized as follows:

(i) Inspired by quantum statistics, we introduce a novel distance between probability distributions called *kernel trace distance* ($d_{KT}$).

(ii) We show that $d_{KT}$ is an IPM and illustrate several of its theoretical properties, mainly: a direct comparison to MMD, robustness to contamination, and statistical convergence rates that do not depend on the dimension.

(iii) We showcase how to compute $d_{KT}$ and illustrate its practical performance on particle gradient flows and Approximate Bayesian Computation (ABC).

**Organisation of the paper** In section 2, we provide some background on quantum statistical distances and introduce $d_{KT}$. In section 3, we explain further the motivation to introduce $d_{KT}$, notably by comparing it with the other distances, MMD in particular. We show in section 4, under some eigenvalue decay rate assumptions, convergence rates that do not depend on the dimension, as well as robustness. In section 2.3, we explain how to compute $d_{KT}$. Finally, we illustrate our findings by experiments in section 5.

## 2 Kernel Trace Distance

For a positive semi-definite kernel $k : \mathcal{X} \times \mathcal{X} \to \mathbb{R}$, its RKHS $\mathcal{H}$ is a Hilbert space of real-valued functions with inner product $\langle \cdot, \cdot \rangle_{\mathcal{H}}$ and norm $\| \cdot \|_{\mathcal{H}}$. It is associated with a feature map $\varphi : \mathcal{X} \to \mathcal{H}$ such that $k(x, y) = \langle \varphi(x), \varphi(y) \rangle_{\mathcal{H}}$. We denote $\mathcal{L}(\mathcal{H})$ the space of bounded linear operators from $\mathcal{H}$ to itself. For a vector $v \in \mathcal{H}$, $v^*$ denotes its dual linear form defined by $v^*(w) = \langle v, w \rangle$ for any

87 $w \in \mathcal{H}$. For an operator $T \in \mathcal{L}(\mathcal{H})$, $T^*$ is its adjoint. $|| \cdot ||_p$ denotes the $p$-Schatten norm explicited
88 below.

**Assumption 0.** In the whole paper, we restrict ourselves to the setting of a completely separable set
90 $\mathcal{X}$, endowed with a Borel $\sigma$-algebra, and a separable RKHS $\mathcal{H}$ of real-valued functions on $\mathcal{X}$, with a
91 bounded continuous strictly positive kernel.

## 2.1 Background

**RKHS density operators [Bach, 2022].** Let $\mu$ a measure on $\mathcal{X}$. Define $\Phi$ the kernel covariance
94 operator embedding as:

$$\Phi : \mu \mapsto \Sigma_\mu = \int_{\mathcal{X}} \varphi(x)\varphi(x)^* d\mu(x). \tag{1}$$

95 We will call $\Sigma_\mu$ the RKHS density operator of $\mu$, in reference to the wording of density operator
96 in quantum information theory: this is to insist that $\Sigma_\mu$ is an embedding in itself (with feature map
97 $\varphi(\cdot)\varphi(\cdot)^*$), rather than just the covariance of a mean embedding with feature map $\varphi$. The operator $\Sigma_\mu$
98 is self-adjoint, and positive semidefinite when $\mu$ is a probability measure. To keep the analogy with
99 quantum density operators, similarly to Bach [2022], we consider kernels respecting the property:

**Assumption 1.** $\forall x \in \mathcal{X}, \ k(x, x) = 1$.

101 to ensure $\operatorname{Tr} \Sigma_\mu = 1$ (as in the sum of all probabilities equals one). If $\forall x \in \mathcal{X}, \ k(x, x) = M$ for a
102 non-zero constant $M \neq 1$, it is will be easy to generalize many of our results later by dividing by
103 $M$, so this assumption is not too restrictive. If the kernel does not verify Assumption 1 but is strictly
104 positive, it is could also be normalised using $\tilde{k}(x, y) = \frac{k(x,y)}{\sqrt{k(x,x)k(y,y)}}$ instead.

**Schatten norms.** We now provide some background on Schatten norms [Simon, 2005]. For an
106 operator $T \in \mathcal{L}(\mathcal{H})$ and $p \in [1, \infty)$, the $p$-Schatten norm is defined as $||T||_p = (\operatorname{Tr}(|T|^p))^{1/p}$ where
107 $|T| = \sqrt{T^*T}$. If $T$ is compact, this can be rewritten as the $p$-vectorial norm of the singular values of
108 $T$. It also admits a dual definition, denoting $q$ such that $1/p + 1/q = 1$:

$$||T||_p = \sup_{U \in \mathcal{L}(\mathcal{H}), ||U||_q = 1} \langle U, T \rangle \tag{2}$$

109 where the inner product is $\langle U, T \rangle = \operatorname{Tr}(U^*T)$.

110 The Schatten 2-norm is the Hilbert-Schmidt norm with respect to this inner product: $||T||_2 = $
111 $\sqrt{\operatorname{Tr}(T^*T)}$. Then, the Schatten $\infty$-norm is the operator norm : $||T||_\infty = \sup_{x \in \mathcal{H} \setminus 0} \frac{||Tx||_{\mathcal{H}}}{||x||_{\mathcal{H}}}$ i.e., the
112 maximum of the singular values of the operator in absolute value. We have the following inequalities:

113 • For $1 \leq p \leq q \leq \infty$: $\forall T \in \mathcal{L}(\mathcal{H}), ||T||_1 \geq ||T||_p \geq ||T||_q \geq ||T||_\infty$.
114 • $\forall T, S \in \mathcal{L}(\mathcal{H}), ||TS||_1 \leq ||T||_2||S||_2$. $\tag{3}$
115 • From this, it can be deduced taking $T$ as the identity operator, for $\mathcal{H}$ of finite dimension:

$$\forall S \in \mathcal{L}(\mathcal{H}), ||S||_1 \leq \sqrt{\dim(\mathcal{H})}||S||_2. \tag{4}$$

## 2.2 Definition

117 In quantum information theory, the trace distance is a mathematical tool that can be used to compare
118 density operators by measuring the Schatten 1-norm of their difference. Inspired by this, we define:

**Definition 2.1.** The *kernel trace distance* between two probability measures $\mu, \nu$ on $\mathcal{X}$ is defined as:

$$d_{KT}(\mu, \nu) = ||\Sigma_\mu - \Sigma_\nu||_1.$$

120 We will also relate it to other distances such as:

• Wasserstein distances [Villani, 2009]:

$$W_d(\mu, \nu) = \inf_{\pi \in \Pi(\mu, \nu)} \iint d(x, y) \mathrm{d}\pi(x, y)$$

122 where $d : \mathcal{X} \times \mathcal{X} \to \mathbb{R}^+$ is a cost and $\Pi(\mu, \nu)$ denotes all the possible couplings between $\mu$
123 and $\nu$. The Wasserstein-$p$ distance is obtained by replacing $d$ by its power $d^p$ in the integral
124 and taking the $p$-root of the whole expression.

- The Bures distance [Bhatia et al., 2019] on positive definite matrices $A$:
$$d_{BW}(A, B) = \sqrt{\operatorname{Tr} A + \operatorname{Tr} B - 2F(A, B)}$$
  where $F(A, B) = \operatorname{Tr}(A^{1/2}BA^{1/2})^{1/2}$ is called the fidelity. It coincides with the Wasserstein-2 distance between two normal distributions (also called Bures-Wassertein distance) with identical mean, and different covariances $A$ and $B$. The formula can be extended to operators with finite traces.

- The Kernel Bures distance [Zhang et al., 2019] is defined as:
$$d_{KBW}(\mu, \nu) = d_{BW}(\Sigma_\mu, \Sigma_\nu).$$

- The Total Variation is a special case of the Wasserstein distance where the cost is $d$ : $(x, y) \mapsto 1_{x=y}$ and can be expressed as:
$$\|\mu - \nu\|_{TV} = \frac{1}{2} \int_{\mathcal{X}} |\mu(x) - \nu(x)| dx$$

- The Maximum Mean Discrepancy [Gretton et al., 2012]:
$$\operatorname{MMD}(\mu, \nu) = \left\| \int_{\mathcal{X}} k(x, \cdot)\mu(x)dx - \int_{\mathcal{X}} k(x, \cdot)\nu(x)dx \right\|_{\mathcal{H}}$$

- Integral Probability Metrics (IPM) [Müller, 1997] defined as:
$$d(\mu, \nu) = \sup_{f \in \mathcal{F}} \{|\mathbb{E}_{X \sim \mu}[f(X)] - \mathbb{E}_{X \sim \nu}[f(X)]|\}$$
  where the function space $\mathcal{F}$ is rich enough to make this expression a metric. The Wasserstein-1 distance, the TV and MMD are IPMs (with $\mathcal{F}$ being 1-Lipschitz functions w.r.t. $\|\cdot\|$, functions with values in [-1,1], and a RKHS unit ball respectively).

**Proposition 2.2.** *If $k^2$ is characteristic i.e $\Phi$ is injective, $d_{KT}$ and $d_{KBW}$ are metrics.*

PROOF. Symmetry, non-negativity, triangle inequality and $d_{KT}(\mu, \mu) = 0$ (resp. $d_{KBW}(\mu, \mu) = 0$) are naturally inherited from the Schatten norm on operators for $d_{KT}$ and from the standard Bures-Wasserstein distance for $d_{KBW}$. Then, as $d_{KT}(\mu, \nu) = 0$ (resp. $d_{KBW}(\mu, \nu) = 0$) implies $\Sigma_\mu = \Sigma_\nu$, injectivity of $\Phi$ enforces $\mu = \nu$.

Examples of characteristic kernels are the family of Gaussian kernels, whose squared kernel also belong to, modulo a change of parameter. On compact set, a sufficient condition for characteristicity is universality [Steinwart, 2001], see for instance Bach [2022].

## 2.3 Computation for discrete measures

As interesting, i.e. expressive RKHS are often of infinite dimension, computations with kernel methods relies on the so-called "kernel trick", reducing computation on the empirical kernel matrix (Gram matrix of two sets of samples using the kernel inner product) which is of finite dimension. It is well-known that the spectrum of the covariance operator $\Sigma_{\mu_n}$ are the ones of the kernel Gram matrix $(k(x_i, x_j))_{i,j=1}^n$ divided by the number of samples [Bach, 2022, Proposition 6]. Here, we generalise the concept for differences of distributions.

First, notice that $\Sigma_{\mu_n} - \Sigma_{\nu_m} = \Sigma_{\mu_n - \nu_m}$, which incites us to consider the samples from each distribution altogether. We denote without duplicates $(z_k)_{k=1,...,r}$ the samples in the union of the sample sets $X, Y$ (corresponding respectively to distributions $\mu_n, \nu_m$), where $r$ is the number of distinct elements in $X, Y$. We note $Z = [\tilde{\varphi}(z_k)]_{k=1...r}$ the column of vectors in $\mathcal{H}$ where $\tilde{\varphi}(z_k) = \sqrt{(\mu_n - \nu_m)(\{z_k\})}\varphi(z_k)$ if $(\mu_n - \nu_m)(\{z_k\}) \geq 0$, $\tilde{\varphi}(z_k) = i\sqrt{|(\mu_n - \nu_m)(\{z_k\})|}\varphi(z_k)$ else.

We can see $Z$ by a slight abuse of notation as the linear map $Z : \mathcal{H} \to \mathbb{C}^r, v \mapsto [\langle\tilde{\varphi}(z_1), v\rangle, ..., \langle\tilde{\varphi}(z_r), v\rangle]$ and by duality $Z^*$ (real not Hermitian adjoint) would be the linear map $Z^* : \mathbb{C}^r \to \mathcal{H}, u \mapsto \sum_{i=1,...,r} u_i\tilde{\varphi}(z_i)$.

Then we define the *difference kernel matrix* as $K = Z^*Z$. Typically, in case where all samples are distinct, $X \cap Y = \emptyset$ and $(\mu_n - \nu_m)(\{z_k\}) = \mu_n(\{z_k\}) = 1/n$ for samples $z_k \in X$ from $\mu_n$ and $(\mu_n - \nu_m)(\{z_k\}) = -\nu_m(\{z_k\}) = 1/m$ for samples $z_k \in Y$ from $\nu_m$, then
$$K = \begin{bmatrix} \frac{1}{n}K_{XX} & \frac{i}{\sqrt{mn}}K_{XY} \\ \frac{i}{\sqrt{mn}}K_{YX} & -\frac{1}{m}K_{YY} \end{bmatrix}$$

where $K_{XX}, K_{YY}, K_{YX}, K_{XY}$ are the usual kernel Gram matrices. Other cases are similar, adjusting the probability weights on rows and columns.

**Proposition 2.3.** *Assume the kernel is such that for any family $(x)$ of distinct elements of $\mathcal{X}$, $(\varphi(x))$ is linearly independent. The difference kernel matrix $K$ as defined just above and $\Sigma_{\mu_n - \nu_m}$ have the same eigenvalues, whose Schatten 1-norm is $d_{KT}(\mu_n, \nu_m)$.*

The proof of Proposition 2.3 is deferred to Appendix A.4. The condition is verified by the Gaussian kernel and more generally it is equivalent to the kernel being strictly positive. It is sufficient to get the eigenvalues by either Autonne-Takagi factorisation [Autonne, 1915, Takagi, 1924], Schur or Singular Value decomposition, and compute their 1-norm. This SVD is of complexity $O(r^3)$ in general.

# 3 Discriminative properties

In this section, we study the discriminative properties of the $d_{KT}$ distance and how it relates to alternative distances between distributions introduced previously.

## 3.1 Comparison with other distances

We first show that our novel distance $d_{KT}$ belongs to the family of Integral Probability Metrics (IPM).

**Proposition 3.1.**

(i) *$d_{KT}$ is an IPM with respect to the function space $\mathcal{F}_1 = \{f : x \mapsto \varphi(x)^* U \varphi(x) | U \in \mathcal{L}(\mathcal{H}), ||U||_\infty = 1\}$.*

*Moreover if Assumption 1 is verified:*

(ii) *functions in $\mathcal{F}_1$ have values in $[-1, 1]$, and*

(iii) *verify the following "Lipschitz" property: $\forall x, y \in \mathcal{X}, |f(x) - f(y)| \leq 2||\varphi(x) - \varphi(y)||_\mathcal{H}$.*

The proof of Proposition 3.1 is deferred to Appendix A.2. Since the TV distance is an IPM with respect to functions bounded by 1, we have the following corollary:

**Corollary 3.2.** $d_{KT}(\mu, \nu) \leq ||\mu - \nu||_{TV}$.

We also have a direct comparison between $d_{KT}$ and a MMD.

**Lemma 3.3.** *The Schatten 2-norm of the difference of the RKHS density operators of two probability distributions $\mu, \nu$ on $\mathcal{X}$ can be identified to their Maximum Mean Discrepancy using the kernel $k^2$:*

$$||\Sigma_\mu - \Sigma_\nu||_2 = \mathrm{MMD}_{k^2}(\mu, \nu)$$

*Consequently, since $d_{KT}$ is a Schatten 1-norm of this difference, $\mathrm{MMD}_{k^2}(\mu, \nu) \leq d_{KT}(\mu, \nu)$.*

This follows mainly from the fact that $\langle \Sigma_\mu, \Sigma_\nu \rangle = \int_\mathcal{X} \int_\mathcal{Y} k(x, y) k(x, y) \mu(x) \nu(y) dx dy$ (see Appendix A.1.1). Finally, we can relate $d_{KT}$ to some Wasserstein distance. Denoting $c_k(x, y) = ||\varphi(x) - \varphi(y)||_\mathcal{H} = \sqrt{2(1 - k(x, y))}$ a cost defined from the kernel $k$, and applying the Lipschitz property of Theorem 3.1, we get the following:

**Corollary 3.4.** *If Assumption 1 is verified, $d_{KT}(\mu, \nu) \leq 2W_{c_k}(\mu, \nu)$. Furthermore, using the Gaussian kernel with parameter $\sigma$,*

$$d_{KT}(\mu, \nu) \leq 2W_{c_k}(\mu, \nu) \leq \frac{2}{\sigma} W_{||.||}(\mu, \nu).$$

The last remark is due to the fact that the Wasserstein-1 distance is an IPM defined by the functions which are 1-Lipschitz w.r.t. $|| \cdot ||$, and for the Gaussian kernel $k(x, y) = e^{-\frac{||x-y||^2}{2\sigma^2}}$, we have $c_k(x, y) \leq \frac{||x-y||}{\sigma}$. See Appendix A.1.1 for full proof.

Finally our novel distance can be related to other kernelized quantum divergences. Some well-known inequality in quantum information theory relating the trace distance and the fidelity is the following Fuchs and Van De Graaf [1999] inequality :

$$2(1 - F(A, B)) \leq ||A - B||_1 \leq 2\sqrt{1 - F(A, B)^2} \tag{5}$$

which translates as upper and lower bounds on $d_{KT}$ with respect to $d_{KBW}$ (see proof in Appendix A.1.1 using Assumption 1):

$$d_{KBW}(\mu, \nu)^2 \leq d_{KT}(\mu, \nu) \leq 2d_{KBW}(\mu, \nu) \tag{6}$$

Let $D_{\mathrm{KL}}(A|B) = \mathrm{Tr}(A(\log A - \log B))$ the quantum relative entropy. The Kernel-Kullback-Leibler (KKL) divergence introduced in Bach [2022] is defined as the latter applied to the density operators of two distributions $\mu, \nu$ on $\mathcal{X}$ (in particular, it is infinite if $\mu$ is not absolutely continuous w.r.t. $\nu$). Thanks to the (quantum) Pinsker's inequality, we have then: $\frac{1}{2}d_{KT}(\mu, \nu)^2 \leq D_{\mathrm{KL}}(\Sigma_\mu|\Sigma_\nu) := \mathrm{KKL}(\mu|\nu)$. Hence, our distance can be framed within several well-known alternative discrepancies.

## 3.2 Normalized energy

From our Assumption 1 on the kernel, we have ensured that for any measure $\mu$, $||\Sigma_\mu||_1 = 1$ which means that all measures representations considered are somehow "normalised". On the contrary, for MMD with $k^2$ (or the Schatten 2-norm), $||\Sigma_\mu||_2$ the "internal energy" depends on the measure (and on the kernel parameters such as bandwidth) and it can be smaller for distributions which are very flat, with high variance, as in general $k(x, y) \leq k(x, x)$ for $x \neq y$. This has consequences as intrinsically $||\Sigma_\mu - \Sigma_\nu||_2 \leq \sqrt{||\Sigma_\mu||_2^2 + ||\Sigma_\nu||_2^2}$, the maximum value can be already small independently of the differences between $\mu$ and $\nu$. When minimizing an objective such as $\mu \mapsto ||\Sigma_\mu - \Sigma_\nu||_2$ (e.g., with gradient descent on the atoms in the support of $\mu$ if it is a discrete measure, as in Arbel et al. [2019]), this has an impact on the shape of the slope. Moreover, the energy depends on

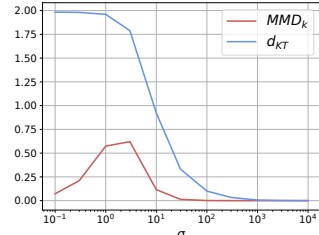

Figure 1: Kernel distances between $\mu = \mathcal{N}(0, 1)$ and $\nu = \mathcal{N}(5, 1)$, as a function of the Gaussian kernel bandwidth $\sigma$.

the hyperparameters of the kernel, which are hard to tune for both the distributions' variances and the distance between their means at the same time.

Figure 1 illustrates this by displaying the two distances between sets of $n = 1000$ samples from $\mathcal{N}(0, 1)$ and $\mathcal{N}(5, 1)$. We would expect sample sets to look closer as the Gaussian kernel bandwidth $\sigma$ grows, but for MMD that is not always the case. Other such phenomena are displayed by varying the variance or the mean of the distributions in the Appendix B.1.

Now let us consider two measures $\mu, \nu$ on $\mathcal{X}$ such that $\mathbb{E}_{X \sim \mu, Y \sim \nu}[k(X, Y)] \leq \epsilon$ for some small parameter $\epsilon > 0$. Then, $\langle \Sigma_\mu, \Sigma_\nu \rangle \leq \epsilon$ by Cauchy-Schwartz. Consider the density operator of the mixture $\Sigma_{\frac{1}{2}\mu + \frac{1}{2}\nu} = \frac{1}{2}\Sigma_\mu + \frac{1}{2}\Sigma_\nu$, we have:

$$||\Sigma_{\frac{1}{2}\mu + \frac{1}{2}\nu}||_1 = 1 = \frac{1}{2}||\Sigma_\mu||_1 + \frac{1}{2}||\Sigma_\nu||_1, \qquad ||\Sigma_{\frac{1}{2}\mu + \frac{1}{2}\nu}||_2^2 \leq \frac{1}{2}\left(\frac{1}{2}||\Sigma_\mu||_2^2 + \frac{1}{2}||\Sigma_\nu||_2^2 + \epsilon\right).$$

We see that in contrast to the 1-Schatten norm, the 2-Schatten norm energy bound is roughly divided by 2 (as $\epsilon \to 0$, e.g. for almost orthogonals $\Sigma_\mu, \Sigma_\nu$). Then, we reason with distance rather than norm:

**Proposition 3.5.** *Let us consider distances between two mixtures $P = \frac{1}{2}\mu_1 + \frac{1}{2}\mu_2$ and $Q = \frac{1}{2}\nu_1 + \frac{1}{2}\nu_2$ such that $\Sigma_{\mu_1}, \Sigma_{\nu_1}$ are orthogonal to $\Sigma_{\mu_2}, \Sigma_{\nu_2}$. Then:*

$$d_{KT}(P, Q) = \frac{1}{2}d_{KT}(\mu_1, \nu_1) + \frac{1}{2}d_{KT}(\mu_2, \nu_2)$$

$$\mathrm{MMD}_{k^2}^2(P, Q) = \frac{1}{4}\mathrm{MMD}_{k^2}^2(\mu_1, \nu_1) + \frac{1}{4}\mathrm{MMD}_{k^2}^2(\mu_2, \nu_2).$$

See proof in the Appendix A.2.1. If the distance between $\mu_2$ and $\nu_2$ are the same as between $\mu_1$ and $\nu_1$ (for instance, if the former are respective translation of the latter and the kernel is translation-invariant), we can see that the squared MMD distance loses a factor 2 while $d_{KT}$ behaves similarly to the Total Variation of the mixtures when $\mu_1, \nu_1$ have different supports than $\mu_2, \nu_2$. This is the case when taking for instance in $\mathcal{X} = \mathbb{R}^2$ $\mu_1 = \mathcal{N}([0, 0], I_2)$ and $\nu_1 = \mathcal{N}([0.3, 0.3], I_2)$ while $\mu_2 = \mathcal{N}(\Delta, I_2)$ and $\nu_2 = \mathcal{N}(\Delta + [0.3, 0.3], I_2)$ for $\Delta = [10, 10]$. In practice, the RKHS density operators are not perfectly orthogonal unless $||\Delta|| \to +\infty$ (in that case $\langle \Sigma_\mu, \Sigma_\nu \rangle \to 0$ for a fixed bandwidth), but typically they can look so up to numerical precision, when using exponentially decreasing kernels (e.g., Gaussian). Taking $n = 100$ samples each from each $\mu_1$ and $\nu_1$, and translating them by $\Delta$, the results above from

Proposition 3.5 are confirmed numerically: we find empirically $\widehat{d_{KT}}(P,Q) = \widehat{d_{KT}}(\mu_1, \nu_1) = 0.5992$ while $\widehat{\text{MMD}}^2_{k^2}(\mu_1, \nu_1) = 0.0253$ but $\widehat{\text{MMD}}^2_{k^2}(P,Q) = 0.0127$, half of it (for a Gaussian kernel with bandwidth $\sigma = 0.5$).

## 3.3 Robustness

We now turn to investigating the robustness of the kernel trace distance. In particular, we consider the $\epsilon$-contamination model, where the training dataset is supposedly contaminated by a fraction $\epsilon \in (0,1)$ of outliers [Huber, 1964]. The following proposition quantifies the robustness of this distance.

**Proposition 3.6.** *Denote $P_\varepsilon = (1-\varepsilon)P + \varepsilon C$ where $C$ is some contamination distribution. We have when Assumption 1 is verified: $|d_{KT}(P_\varepsilon, Q) - d_{KT}(P,Q)| \leq 2\varepsilon$.*

The proof relies on the triangular inequality (see Appendix A.3.2). Hence, we see that $d_{KT}$ is robust while for the Wasserstein distance, a contamination $C$ arbitrarily "far away from the distribution $Q$" will incur an arbitrarily high distance. The proof of robustness also works for MMD.

# 4 Statistical Properties

## 4.1 Convergence rate

In this section, we consider a measure $\mu$ and its empirical counterpart $\mu_n$ for $n$ independent samples and study the rate of convergence of $d_{KT}(\mu, \mu_n)$. We note $A \lesssim_{\mu^{\otimes n}} b$ where $A$ is r.v., when for any $\delta > 0$, there exists $c_\delta < \infty$ such that $\mu^{\otimes n}(A \leq c_\delta b) \geq \delta$. With the Schatten 1-norm, it is not enough to study only the concentration of one (the maximal) eigenvalue as for the operator norm ($p = \infty$), we need to handle an infinity of eigenvalues (when the RKHS is of infinite dimension), neither can we use the Cauchy-Schwarz trick as for the Hilbert norm ($p = 2$). However, since the trace of our kernel density operators are bounded by 1, only a few of the eigenvalues will have a significant contribution. Therefore, assuming some decay rate on those eigenvalues, we can focus on the convergence of operators on a subspace of the top eigenvectors, using results from the Kernel PCA literature. We introduce the population and empirical square loss associated with some projector $P$:

$$R(P) = \mathbb{E}_{X \sim \mu}||\phi(X) - P\phi(X)||^2_{\mathcal{H}}, \qquad R_n(P) = \sum_{i=1}^n \frac{1}{n}||\phi(x_i) - P\phi(x_i)||^2_{\mathcal{H}}$$

where the $(x_i)_{i=1...n}$ are each drawn independently from $\mu$. We first make the following assumption, as in Sterge et al. [2020].

**Assumption 2.** *The eigenvalues $(\lambda_i)_{i \in I}$ of $\Sigma_\mu$ (resp. $(\hat{\lambda}_j)_{j \in J}$ of $\Sigma_{\mu_n}$) are positive, simple and w.l.o.g. arranged in decreasing order ($\lambda_1 \geq \lambda_2 \geq ...$).*

This allows us to denote $P^l(\Sigma_\mu)$ the projector on the subspace of the $l$ eigenvectors associated with the $l$ highest eigenvalues $\lambda_1, ..., \lambda_l$. Note that $||P^l(\Sigma_\mu)\Sigma_\mu - \Sigma_\mu||_1 = \sum_{i>l} \lambda_i = R(P^l(\Sigma_\mu))$ (see for instance Blanchard et al. [2007], Rudi et al. [2013]). Similarly we consider $P^l(\Sigma_{\mu_n})$ for $\Sigma_{\mu_n}$.

We now consider different kinds of assumptions on the decay rate of eigenvalues of $\Sigma_\mu$ to get different corresponding convergence rates, as in Sterge et al. [2020], Sterge and Sriperumbudur [2022].

**Assumption P (Polynomial).** *For some $\alpha > 1$ and $0 < \underline{A} < \bar{A} < \infty$,*

$$\underline{A}i^{-\alpha} \leq \lambda_i \leq \bar{A}i^{-\alpha}. \tag{P}$$

**Assumption E (Exponential).** *For $\tau > 0$ and $\underline{B}, \bar{B} \in (0, \infty)$,*

$$\underline{B}e^{-\tau i} \leq \lambda_i \leq \bar{B}e^{-\tau i}. \tag{E}$$

**Lemma 4.1.** *Suppose Assumption 1 and 2 are verified. With a polynomial decay rate of order $\alpha > 1$ (Assumption P), for $l = n^{\frac{\theta}{\alpha}}, 0 < \theta \leq \alpha$:*

$$||P^l(\Sigma_\mu)\Sigma_\mu - \Sigma_\mu||_1 = R(P^l(\Sigma_\mu)) = \Theta\left(n^{-\theta(1-\frac{1}{\alpha})}\right), \quad ||P^l(\Sigma_\mu)\Sigma_\mu - \Sigma_\mu||_2 = \Theta\left(n^{-\theta(1-\frac{1}{2\alpha})}\right), \tag{7}$$

*and there exists $N \in \mathbb{N}$ such that for $n > N$:*

$$||P^l(\Sigma_{\mu_n})\Sigma_\mu - \Sigma_\mu||_2 \lesssim_{\mu^{\otimes n}} max(n^{-\frac{1}{2}+\frac{1}{4\alpha}}, n^{-\theta+\frac{1}{4\alpha}}). \tag{8}$$

289 *With an exponential decay rate (Assumption E), for $l = \frac{1}{\tau}\log n^\theta, \theta > 0$:*

$$||P^l(\Sigma_\mu)\Sigma_\mu - \Sigma_\mu||_1 = R(P^l(\Sigma_\mu)) = \Theta(n^{-\theta}), \qquad ||P^l(\Sigma_\mu)\Sigma_\mu - \Sigma_\mu||_2 = \Theta\left(n^{-\theta}\right) \quad (9)$$

290 *and there exists $N \in \mathbb{N}$ such that for $n > N$:*

$$||P^l(\Sigma_{\mu_n})\Sigma_\mu - \Sigma_\mu||_2 \lesssim_{\mu^{\otimes n}} \begin{cases} \sqrt{\frac{\log n}{n^\theta}} & \text{if } \theta < 1 \\ \frac{(\log n)}{\sqrt{n}} & \text{if } \theta \geq 1. \end{cases} \quad (10)$$

291 The previous lemma (see proof in Appendix A.3.1) is crucial to prove our main theorem below, that
292 provides dimension-independent statistical rates.

293 **Theorem 4.2.** *Suppose Assumption 1 and 2 are verified.*

294 • *If the eigenvalues of $\Sigma_\mu$ follow a polynomial decay rate of order $\alpha > 1$ (Assumption P),*
295 *then:*

$$d_{KT}(\mu, \mu_n) \lesssim_{\mu^{\otimes n}} n^{-\frac{1}{2} + \frac{1}{2\alpha}}.$$

296 • *If the eigenvalues of $\Sigma_\mu$ follow an exponential decay rate (Assumption E), then:*

$$d_{KT}(\mu, \mu_n) \lesssim_{\mu^{\otimes n}} \frac{(\log n)^{\frac{3}{2}}}{\sqrt{n}}.$$

297 SKETCH OF PROOF.   For clarity of notation, we abbreviate $\Sigma_\mu$ and $\Sigma_{\mu_n}$ as $\Sigma$ and $\Sigma_n$. By the
298 triangular inequality:

$$||\Sigma - \Sigma_n||_1 \leq ||\Sigma - P^l(\Sigma)\Sigma||_1 + ||(P^l(\Sigma) - P^l(\Sigma_n))\Sigma||_1 + ||P^l(\Sigma_n)(\Sigma - \Sigma_n)||_1$$
$$+ ||P^l(\Sigma_n)\Sigma_n - \Sigma_n||_1 \coloneqq (A) + (B) + (C) + (D) \quad (11)$$

299 We bound each term of eq. 11. Term (A) is bounded using Lemma 4.1. Similarly, (D) relates to (A)
300 by a result due to Blanchard et al. [2007] (eq. (30)), see Lemma A.3 in Appendix A.3.1. For (B) and
301 (C), the projections allow to work in a subspace of dimension at most $2l$ and by eq. (3) (Hölder's
302 inequality) to relate to the Schatten 2-norm which has rates like MMD. Finally, we pick $\theta = \frac{1}{2}$ for
303 polynomial decay and $\theta = 1$ for the exponential decay (see Lemma 4.1) to minimise the maximum
304 of the four terms. See Appendix A.3.1 for the full proof.

305 By the Fuchs-van de Graaf inequality (Eq. (5) and (6)), it directly implies (also dimensionally-
306 independent) convergence rates for the Kernel Bures Wasserstein distance, that are novel to the best
307 of our knowledge.

308 **Corollary 4.3.** *Suppose Assumption 1 and 2 verified.   If Assumption P is verified:*
309 $d_{KBW}(\mu, \mu_n) \lesssim_{\mu^{\otimes n}} n^{-\frac{1}{4} + \frac{1}{4\alpha}}$. *If Assumption E is verified:* $d_{KBW}(\mu, \mu_n) \lesssim_{\mu^{\otimes n}} (\log n)^{\frac{3}{4}} n^{-\frac{1}{4}}$.

## 310 5   Experiments

311 In this section, we illustrate the interest of our novel kernel trace distance on different experiments.

312 **Approximate Bayesian Computation (ABC)**   The purpose of Approximate Bayesian Compu-
313 tation [Tavaré et al., 1997] is to compute an approximation of the posterior when doing Bayesian
314 inference in a likelihood-free fashion. The idea of using a distance $d$ between distributions to build a
315 synthetic likelihood has recently flourished [Frazier, 2020, Bernton et al., 2019, Jiang, 2018]. ABC
316 methods based on IPM enjoy theoretical guarantees [Legramanti et al., 2022]. The ABC posterior
317 distribution is defined by $\pi(\theta|X^n) \propto \int \pi(\theta)\mathbb{1}_{\{d(X^n, Y^m) < \epsilon\}} p_\theta(Y^m)\mathrm{d}Y^m$, where $\pi(\theta)$ is a prior over
318 the parameter space $\Theta$, $\epsilon > 0$ is a tolerance threshold, and $Y^m$ are synthetic data generated according
319 to $p_\theta(Y^m) = \prod_{j=1}^m p_\theta(Y_j)$. It is approximately computed by drawing $\theta_i \sim \pi$ for $i = 1, ..., T$ and
320 simulating synthetic data $Y^m \sim p_{\theta_i}$ and keeping or rejecting $\theta_i$ according to whether the synthetic
321 data is close to the real data. The result is a list $L_\theta$ of all accepted $\theta_i$ (see Algo. 1 in the Appendix B.2).

322 Here, as we are interested in robustness, we will consider a contamination case using Normal
323 distributions but where nonetheless the usual likelihood fails to recover the correct mean as the data is
324 corrupted. We will take as prior $\pi = \mathcal{N}(0, \sigma_0^2)$ and the real data consist of $n = 100$ samples coming
325 following $\mu^* = \mathcal{N}(\theta^* = 1, 1)$ where 10% of the samples are replaced by contaminations from
326 $\mathcal{N}(20, 1)$. We fit the model $p_\theta = \mathcal{N}(\theta, 1)$ by picking the best $\theta$ possible. We carry out $T = 10000$
327 iterations, generating each times $m = n$ synthetic data.

We consider ABC with the threshold value $\epsilon = 0.05, 0.25, 0.5, 1$. For the proposed distance $d_{KT}$, Bayes' rule gives posterior $p(\theta|x) = \mathcal{N}(\frac{\sum_{i=1}^{n} x_i}{n + \frac{1}{\sigma_0^2}}, \frac{1}{n + \frac{1}{\sigma_0^2}})$. Since $\mathbb{E}[X_i] = 0.9 \times 1 + 0.1 \times 20 = 2.9$ the location is therefore in expectation $\mathbb{E}[\frac{\sum_{i=1}^{n} x_i}{n + \frac{1}{\sigma_0^2}}] = \frac{n}{n + \frac{1}{\sigma_0^2}} 2.9 \approx 2.9$, the contamination significantly impacted the posterior. Similarly, for any model $p_\theta$, the Wasserstein distance with the contaminated mixture $0.9\mathcal{N}(1, 1) + 0.1\mathcal{N}(20, 1)$ will be high, and empirically all of the $T$ iterations are rejected for all the values of $\epsilon$ considered. Thus, we disregard the Wasserstein distance from the experiment and compare the performance of MMD to that of $d_{KT}$. We also consider concurrent methods out of our scope such as MMD with the unbounded energy kernel: $k(x, y) = \frac{1}{2}(||x|| + ||y|| - ||x - y||)$ Sejdinovic et al. [2013], and others displayed in Appendix B.2.

We measure the average Mean Square Error between the target parameter $\theta^* = 1$ and the accepted $\theta_i \in L_\theta$: $\widehat{MSE} = \frac{1}{|L_\theta|} \sum_{\theta_i \in L_\theta} ||\theta_i - \theta^*||^2$ which also corresponds to the average of squared Wasserstein 2-distance as $W_2^2(\mu^*, p_{\theta_i}) = ||\theta_i - \theta^*||^2$ since we consider only Gaussians with same variance. We picked $\sigma_0 = 5$ for the prior. We repeat 10 times the experiment with fresh samples, the averaged results are shown in Table 1. As expected – and discussed in subsection 3.2 – MMD (gaussian) is too lenient to accept. For $\epsilon = 0.05$ inferior to the contamination level (10%), it still accept 11% of the times, while $d_{KT}$ reject all the times, which can be understood as $d_{KT}$ detecting the contamination, that prevents to match with the Gaussian model. The energy kernel can not help enough to beat $d_{KT}$. The densities of the obtained posteriors are shown alongside the target in Fig. (4) and (5) in the Appendix B.2.

Table 1: Average MSE of ABC Results.
The Gaussian kernel is used with $\sigma = 1$ (as the variance of $p_\theta$ and $\mu^*$). As expected, MMD is too lenient to accept most sampled $\theta_i$ leading to a high average MSE unless $\varepsilon$ is carefully chosen. Whereas the proposed $d_{KT}$ discriminates between the correct and the wrong $\theta_i$ for $\varepsilon$ larger than the contamination threshold 0.1. MMD is assumed to use the Gaussian kernel while $\mathrm{MMD_E}$ denotes the MMD with the energy kernel.

| $\varepsilon$ | 0.05 | | | 0.25 | | | 0.5 | | |
|---|---|---|---|---|---|---|---|---|---|
| distance | MMD | $\mathrm{MMD_E}$ | $d_{KT}$ | MMD | $\mathrm{MMD_E}$ | $d_{KT}$ | MMD | $\mathrm{MMD_E}$ | $d_{KT}$ |
| #accept. | 1092 | 0 | 0 | 2964 | 0 | 58 | 6168 | 846 | 828 |
| MSE | 0.19 | N/A | N/A | 1.29 | N/A | **0.03** | 7.47 | 0.17 | 0.12 |

**Particle Flow** We consider the performance of gradient descent when optimizing $\mu \mapsto d_{KT}(\mu, \nu)$ for discrete measures $\mu, \nu$ on $\mathbb{R}^2$, given an initial point cloud (in red) and a target cloud of points (in blue) both of $n = 100$ points. We run the scheme with a learning rate of 0.005 for 1000 steps, using $d_{KT}$ (Schatten 1-norm) and MMD (Schatten 2-norm), see Appendix B.3 (Figs. 6 and 7). We use the Laplacian kernel: $k(x, y) = e^{-\frac{||x-y||_1}{\sigma}}$ where here $|| \cdot ||_1$ means the $l_1$ norm for vectors. We choose a bandwidth $\sigma = 1$ (as the image size is a unit square) for $d_{KT}$ and for MMD we use $k^2$ as kernel to match the Schatten 2-norm (i.e. we use $\sigma = 0.5$ instead of $\sigma = 1$, and it gives a better convergence). The inherent internal energy of MMD incites the point cloud to spread out and therefore some particles are still left out far away from the target, which does not happen with $d_{KT}$.

# 6 Conclusion

We introduced a robust distance between probability measures, based on RKHS density (or covariance) operators and their Schatten-1 norm. It is the greatest in a family of kernel-based IPM including MMD, and so is more discriminative as shown in experiments. We show how to compute it between discrete measures via a new kernel trick. Assuming some decay rate of the eigenvalues of the RKHS density operator leads to a statistical convergence rate that can be close to $O(n^{-\frac{1}{2}})$. This implies the first (dimension-independent) rates for the Kernel Bures Wasserstein distance. Future work includes reducing computational complexity via Nyström method, improving the dependence on the order of decay $\alpha$, as well as minimax lower bounds.

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
