# OpenReview forum: "Kernel Trace Distance: Quantum Statistical Metric between Measures through RKHS Density Operators"
_NeurIPS.cc/2025/Conference — Submitted to NeurIPS 2025_

### Official Review · Reviewer_B5gM · 2025-06-06

**Clarity:** 4
**Significance:** 4
**Originality:** 4
**Rating:** 5
**Confidence:** 4

**Summary:**

The authors propose a novel distance between probability distributions called kernel trace distance and demonstrates that it is an Integral Probability Metric. The new distance measure is also robust to contamination and has statistical convergence rates that do not depend on the dimension. The authors demonstrate the practicality on particle gradient flows and Approximate Bayesian Computation.

The kernel trace distance is defined as the Schatten 1-norm (the sum of the singular values) of the difference between the density operators corresponding to each probability distribution (i.e. the difference between the sums of the singuar values of each density operator). The density operator is associated with the given kernel covariance operator which generalized the classical covariance matrix to infinite-dimensional feature space.

Because the density operator is infinite-dimensional, computing the singular values of these operators is not tractable. Drawing inspirations from the original kernel trick, the authors propose to take samples from both probability distributions in consideration. The union of the samples is denoted as $(z_k)_{k = 1, \cdots, r}$. Assuming all samples are distinct (which ensures linear independence of the samples), the authors derive a novel kernel trick for deriving the difference kernel matrix. The claim is that the singular values of this alternate difference kernel matrix can be used to compute the singular values of the difference of the density operators; the authors provide a proof.

The authors also prove nice properties of the proposed distance metrics: 1) the proposed distance is an IPM; 2) the proposed distance (the Schatten 1-norm of the difference) lower bounds the Schatten 2-norm of the difference of the density operators; 3) the proposed distance is upper bounded by a related Wasserstein distance; 4) the proposed distance is related to other kernelized quantum divergences; 5) the proposed distance is more discriminative. This is shown in Figure 1 which demonstrates that the increase in the bandwidth parameter should decrease the difference to zero but this is more pronounced in the proposed distance and the MMD does have a peak in the middle. The authors extend this to analysis in which the distance between two mixture distributions are considered and demonstrate that the proposed metric is more discriminative (Proposition 3.5); 6) the proposed distance is more robust. An $\epsilon$ contmination does not stray too much from the true distance, whereas the similar contamination can impact distances such as the Wasserstein distance.

Lastly, the authors discuss convergence. The first part is the convergence when considering that the eigenvalues are truncated to the top ones. The second is due to the approximation of the density operator with a finite number of samples from both probability distributions.

**Questions:**

1. Are there some principled ways of choosing the hyperparameters of the kernel used in the experiment?
2. Compared to the other distance metrics, do you see any tradeoff in terms of incurred computational requirement, other than the standard cost required for SVD computation on the difference kernel matrix?
3. Do the authors see any sparsity tricks that could be applied for the difference kernel matrix? How would this change the properties of the distance metric?
4. I did find Section 3.2 a bit long and the part of the reason is that it combines the discussion between distance between two unimodal Gaussians and the distance between two mixture distributions. There needs to be some pause between these topics.
5. Do the author have the reference for the proof of robustness for MMD (Line 262)?
6. Do the author have plans for verifying some of the assumptions used in analysis of the convergence rate section, for example, Assumption P and E on some datasets?

**Ethical Concerns:**

["NO or VERY MINOR ethics concerns only"]

**Final Justification:**

The authors' responses are satisfactory in addressing my concerns and they have agreed to revise the document for more clarity. The proposed distance measure is positioned nicely along with the currently used ones and the paper is recommended for acceptance.

**Limitations:**

There are no significant limitations. I would encourage the authors to pursue a follow-up paper on the application because there are so many directions you can take this, such as using the distance in generative neural network, etc. This is an interesting submission with many applications.

**Paper Formatting Concerns:**

There are no major formatting concerns.

**Quality:**

4

**Strengths And Weaknesses:**

Strengths: I think this manuscript is well-written and the authors did make considerable efforts to make rather theoretical aspects accessible to practitioners in kernel methods. The flow of the paper is logical, starting from discussing the proposed distance in relation to the distance metrics that are already in use and proving nice properties such as discriminative power and convergence. The authors derive a new kernel trick which I think is very interesting. I did go through some of the proofs, both in the main writeup and appendix and they seem to be correct.

Weaknesses: the paper is largely theoretical (with significant results!) so I think rather limited results in experimental section can be understandable. One small complaint in the experiment section is the choice of the parameter (such as using the Gaussian kernel with a fixed bandwidth). I will also make additional comments regarding this in the question section.

Presentation-wise: there are some things that could be highlighted such as the discriminative power of the kernel, e.g. Figure 1 which I think is an interesting figure should deserve more attention. But from casual reading, this is not easy to see (I was just able to gather this from my third pass over the paper).

---

> ### Author Rebuttal · Authors · 2025-07-31
>
> We sincerely thank the reviewer for their thorough and thoughtful review and his very positive comments on our paper. We answer his questions below.
>
> 1. We chose the parameter $\sigma = 1$ in both experiments for practical reasons: in the particle flow setting, this value matched the scale of the image data. While we could have used the median heuristic (i.e., the "median trick," based on pairwise distances), it yielded a similar value. In the ABC experiment, $\sigma = 1$ was also natural, as the models involved Gaussian distributions with standard deviation one.
>
> 2.  See answer to Q1/Q2 to Reviewer vtMw.
>
> 3. Building on the previous point, there also exist methods to approximate eigenvalues more efficiently. Since our analysis assumes that only the top eigenvalues are most relevant, such approximations could be a valid way to accelerate computation, though they would reduce the precision of the distance estimate and weaken theoretical guarantees.
>
> 4. We grouped the two experiments together because the first provides intuition that helps in understanding the second, more complex case. However we agree with the reviewer suggestions and have reorganized this section.
>
> 5. The proof for MMD follows a similar structure: it begins and ends in the same way, with the key step using the inequality between the 2-norm and the 1-norm. See also the alternative proof provided by reviewer vtMw.
>
> 6. See answer to first Question of Reviewer MQLz.

---

> > ### Comment · Reviewer_B5gM · 2025-08-02
> > **Dear authors**
> >
> > I am satisfied with your responses to the questions and currently in favor of acceptance. I am looking forward to engaging in further discussion with the other reviewers.

---

### Official Review · Reviewer_MQLz · 2025-06-26

**Clarity:** 4
**Significance:** 3
**Originality:** 3
**Rating:** 5
**Confidence:** 3

**Summary:**

This paper studies a new distance between probability measures defined using a Schatten norm on the kernel covariance operators. This novel distance has advantages over MMD while not suffering from the curse of dimensionality present for Wasserstein distances. Theoretical properties of the distance are studied, including its robustness and efficient estimation. Experiments on Approximate Bayesian Computation and Particle Flow are given.

**Questions:**

- How restrictive are the assumptions on the decay rate of eigenvalues of the operators that are used in the statistical rate proofs?
- How do different choices of $p$ in the Schatten norm affect the kernel trace distance?
- Are there other estimators of this distance that one can define using, for example, the IPM formulation and neural networks?
- Can the authors offer more guidance on how to use the theory developed here for choosing this distance over other existing distances in the zoo of metrics over probability measures?

**Ethical Concerns:**

["NO or VERY MINOR ethics concerns only"]

**Final Justification:**

The authors have adequately addressed my questions, and I feel that this is a valuable addition to the zoo of distances between probability distributions. Therefore I have updated my score to recommend acceptance.

**Limitations:**

yes

**Quality:**

3

**Strengths And Weaknesses:**

Strengths:

- The newly formulated distance strikes a balance between MMD and Wasserstein distance. In particular, it is efficient to estimate, robust, and has an integral probability formulation.
- This work contributes nicely to a growing family of probability metrics defined on kernel covariance operators.

Weaknesses:

- In the zoo of distances, while this has some intriguing properties, it is hard to get a full picture of when to use this distance versus when not to. Some comparisons are given, but they are not informative about what settings this is most useful for.
- The experiments are light and somewhat glossed over. While they show advantages for the proposed distance, they are not extensive enough to explore the advantages/disadvantages of the proposed distance.
- The particle flow experiment is mentioned in the main body but no qualitative/quantitative results are given in the main body of the text.

---

> ### Author Rebuttal · Authors · 2025-07-31
>
> We thank the reviewer for his careful review and positive comments about our paper. We address his main concerns and answer questions below.
>
> **Weaknesses**
>
> - see Answer to Q1/Q2 of Reviewer vtMw: our study show the clear statistical and geometrical benefits of using $d_{KT}$ distance over $MMD$, both theoretically and experimentally. Yet, the computational cost of using our distance is higher than MMD, so this reflects a tradeoff.
>
> - empirical limitations: please see our answer to Q4-5-6 of Reviewer S4kf.
>
> - particle flow: we will add a brief summary of the particle flow results in the main text to complement the details provided in the appendix B.3. Thank you for the suggestion.
>
> **Questions**
>
> - In practice, the assumptions on eigenvalue decay are not overly restrictive. There are known connections between empirical and population-level spectral decay, which make it possible to validate the assumptions through simulation. For example, even a uniform distribution over a cube exhibits polynomial decay when using the Gaussian kernel—despite the fact that the Wasserstein distance suffers from the curse of dimensionality in such settings. Similar behavior holds for other kernel-distribution pairs. For further discussion and concrete examples (including closed-form cases), we refer to Francis Bach’s blog series “Unraveling spectral properties of kernel matrices”, Parts I and II. We will add this discussion to the revised version of the manuscript.
>
> - The Trace Distance corresponds to the case $p = 1$, while MMD with kernel $k^2$ is equivalent to the case $p = 2$. As $p$ increases, the space of discriminating functions in the IPM framework becomes smaller, while the associated convergence rates improve.
>
>
> - One possible extension would be to learn the feature map using a neural network, as in the work of Mroueh et al. (2017), cited in the introduction. This could be done by training on a subset of the data or using an unsupervised approach. In that case, the RKHS would be finite-dimensional, and the kernel could be computed directly as $k(x, y) = \langle \varphi(x), \varphi(y) \rangle$, where $\varphi$ denotes the learned feature map. The difference kernel matrix and its $p$-Schatten norm could then be evaluated accordingly. However, it is important to note that the learned feature map must capture information relevant to both distributions simultaneously, which is not guaranteed if the training is performed using only one of them.
>
>
> - In our work, we focused on MMD, the Wasserstein distance, and the Kernel Trace distance. In particular, Lemma 3.3 and Corollary 3.4 (for the Gaussian kernel) establish the inequality:
> $$MMD_{k^2}(\mu, \nu) \leq d_{KT}(\mu, \nu) \leq \frac{2}{\sigma} W_{\|\cdot\|}(\mu, \nu).$$
> All three are instances of Integral Probability Metrics (IPMs). When choosing among them, one useful guideline is that moving higher in this hierarchy yields more discriminative power, but at the cost of slower statistical convergence and greater computational complexity.

---

### Official Review · Reviewer_S4kf · 2025-06-30

**Clarity:** 2
**Significance:** 2
**Originality:** 2
**Rating:** 3
**Confidence:** 3

**Summary:**

This paper proposes a new distance between probability distributions based on the Schatten norm of their kernel covariance operators. It lies between MMD and Wasserstein distance, offering better discriminative power and robustness to hyperparameters. The method retains the efficiency of kernel-based approaches, helping avoid the curse of dimensionality. Practical computation is enabled via a new kernel matrix formulation, with applications shown in Bayesian inference and particle simulations.

**Questions:**

NO

**Ethical Concerns:**

["NO or VERY MINOR ethics concerns only"]

**Final Justification:**

I sincerely thank the authors for their detailed responses, which address some of my initial concerns. I have also read the comments from other reviewers and acknowledge the theoretical contributions of the paper. I have decided to revise my score to reflect this. However, I remain concerned that the experimental results are not fully convincing, despite the authors' explanations regarding my previous questions. I explain my concerns in detail as follows.

1. **Experimental results in Table 1:**
   Table 1 compares MMD and the proposed $d_{KT}$ under different values of $\epsilon$. At $\epsilon = 0.05$, $d_{KT}$ rejects all samples and ABC fails, and MMD accepts 1,092 samples with an ABC MSE of 0.19. At $\epsilon = 0.25$, $d_{KT}$ achieves an MSE of 0.03 and accepts 58 samples, and MMD accepts more samples (2,964) with an MSE of 1.29. At $\epsilon = 0.5$, MMD accepts 6,168 samples with an MSE of 7.47, and $d_{KT}$ accepts 828 samples with an MSE of 0.12. We can observe that both MMD and $d_{KT}$ accept fewer samples as $\epsilon$ decreases. Therefore, for small $\epsilon < 0.05$, MMD may still perform well and achieve a very low MSE (e.g., 0.03 or <0.03) when accepting few samples (e.g., 58 samples or fewer), while $d_{KT}$ fails, as shown in the case where $d_{KT}$ rejects all samples and ABC fails at $\epsilon = 0.05$. In such cases, MMD can perform over a larger range of $\epsilon$ (including $\epsilon < 0.25$), while $d_{KT}$ performs well only within a narrower range. Thus, MMD is more robust in such cases. Furthermore, to achieve a small MSE, both MMD and $d_{KT}$ require careful tuning of $\epsilon$.

2. **Limited comparison across relevant settings:**
   The authors do not compare MMD and $d_{KT}$ in the two-sample testing setting, where MMD was originally proposed. Similarly, while the KT metric was first introduced by Mroueh et al. [2017] for evaluating generative models, the authors do not evaluate $d_{KT}$ against the original KT metric in that context either. Instead, they focus on the ABC setting and highlight the advantages of $d_{KT}$ over MMD in this specific application (as noted, "particularly relevant in applications such as particle flows ABC"). I think this comparison may not be entirely fair, and a comparison between $d_{KT}$ and distance metrics specifically designed for ABC would be more appropriate.

**Limitations:**

YES

**Quality:**

2

**Strengths And Weaknesses:**

**Strengths**
1. Unlike prior methods that rely on approximations, the proposed approach leverages kernel features in the dual space to derive a closed-form expression for the metric using the kernel trick, enhancing both theoretical clarity and computational efficiency.
2. The authors provide formal guarantees for their proposed distance and demonstrate its utility across a range of numerical applications beyond GANs, indicating greater generality and robustness.
3. The proposed distance $d_{\text{KT}}$ is shown to be an integral probability metric (IPM) with desirable theoretical properties, including robustness to contamination and dimension-independent convergence rates, making it well-suited for high-dimensional and noisy data settings.

**Weaknesses**
1. As stated in lines 58–67, the proposed metric is the same as that in Mroueh et al. [2017], but the authors use kernel features directly in the dual formulation and derive a closed-form expression for the metric by leveraging a kernel trick. Furthermore, the motivation for using this metric differs from previous approaches, as stated in lines 38–40, which note that MMD has several pitfalls due to the different energies of mean embeddings. However, these pitfalls are not clearly explained, and it remains unclear why the proposed metric can effectively avoid them.
2. The notation $k^2$, which first appears in Proposition 2.2, is not clearly defined.
3. The advantage of the KT metric over MMD appears to stem from its "normalized" formulation. While MMD accounts for the internal energy associated with the variance of the distributions, doesn't this suggest that the internal energy form may better reflect the distribution and more effectively distinguish between two distributions with the same parameter?
4. the theoretical analysis mainly compare the value of different metrics, however in statistical testing, how large is the value is not that important, since we always compare the value of test statistic with a testing threshold, normally, when the test satistic have larger scale the testing thresholds will be larger. To compare MMD and the KT , it is important to analyse how they performs compared to testing thresholds.
5. MMD was originally proposed for the two-sample testing task, so more comparisons between KT and MMD within this context would be appreciated. Similarly, the KT metric was first introduced by Mroueh et al. [2017] for evaluating generative models, and additional experiments in this setting would also be valuable.
6. The experiment in Table 1 does not clearly demonstrate the advantages of KT. For smaller values of $\epsilon$, KT appears to underperform, while MMD yields better results. In contrast, for larger $\epsilon$, KT shows improved performance, whereas MMD results in higher MSE. More experiments with varying values of $\epsilon$ would be appreciated.

---

> ### Author Rebuttal · Authors · 2025-07-31
>
> We thank the reviewer for his constructive review, and discuss his concerns below.
>
>
> 1-3: We address these pitfalls of MMD more fully in Section 3.2 and Appendix B.1. Let us clarify our findings afterwards.
> Section 3.2 of the paper introduces the concept of normalized energy in the context of comparing probability measures via RKHS-based operators. We highlight a key distinction between our proposed Kernel Trace (KT) distance and the widely used Maximum Mean Discrepancy (MMD), that both a covariance mean embedding. The information about the distribution is contained in the eigenvalues (similar to probability mass) and eigenvectors (similar to distribution support) of the RKHS density operator. The MMD relies on the Schatten 2-norm and is sensitive to the internal "energy"
> $$||\Sigma_\mu|| = \int\int k(x,y)^2 \mu(x)  \mu(y)dx dy = \int_\mathcal{X} \int_\mathcal{Y} e^{-||x-y||^2/\sigma^2} \mu(x)  \mu(y)dx dy$$, (where $||\cdot||$ denotes the 2-norm here), which is itself related to the variance (or "spread") of distributions. It depends not only on the distribution but also on the kernel's bandwidth. Since our focus is on the distance between two embeddings—rather than representing a single embedding—we showed in Section 3.2 that embeddings with low energy can yield a low MMD value even when they differ significantly. This is because the bandwidth parameter cannot be simultaneously tuned to capture both scale and difference effectively.
> In contrast, the $d_{KT}$ distance uses the Schatten 1-norm, which equals one for all distribution (for all $\mu$, $|\Sigma_{\mu}| = 1$ where $|\cdot|$ denotes the 1-norm here) since we assumed the trace of RKHS density operators is 1. This normalization ensures that the KT distance focuses purely on the separation between distributions, not on their individual energy levels. Our empirical results (e.g., Figure 1) show that KT distance behaves better in this sense than MMD when comparing two fixed distributions while increasing the kernel bandwidths:
> $d_{KT}$ decreases as expected, while MMD is not monotone. In Figure 2, one can see that the $d_{KT}$ distance attains its theoretical maximal value (2), while MMD achieves a much smaller value (value 2 would be approachable by two far-away Dirac). In Figure 3, our distance drops below its maximum value of 2 only when the supports begin to significantly overlap (i.e. when the std deviation $s$ is of the order of the distance between the mean); in contrast, MMD decreases much earlier, as it naturally diminishes with increasing variance. Furthermore, as a second step, we demonstrate (Proposition 3.5) that for mixtures of distributions, the $d_{\mathrm{KT}}$ distance scales linearly with the component-wise distances, thereby preserving discriminability, whereas MMD tends to lose resolution (by a factor 2) due to the averaging effect inherent in its energy-based formulation. Generally, this section is intended to justify why $d_{KT}$ leads to a more stable and robust behavior, making it more reliable in scenarios where MMD’s performance deteriorates. We will highlight those clarifications in paper.
>
>
> 2: $k^2$ is just the squared function of the kernel function $k$ (whose general notation $k : \mathcal{X} \times \mathcal{X} \to \mathbb{R}$ as can be found as the very beginning of section 2), therefore $k^2 : \mathcal{X} \times \mathcal{X} \to \mathbb{R}, (x,y) \mapsto k(x,y)^2$ (positive definiteness is carried on by squaring). Notice that  $$ \langle \varphi(x) \otimes \varphi(x), \varphi(y) \otimes \varphi(y) \rangle = \langle \varphi(x), \varphi(y)  \rangle \langle \varphi(x), \varphi(y)  \rangle = k(x,y)^2$$.
>
>
> 4. We agree with the reviewer’s observation that, in statistical testing, the scale of a test statistic alone does not determine its effectiveness; rather, it should be evaluated in relation to the corresponding testing threshold under the null distribution. However, the utility of a distance metric is not limited to its discriminative power near the null—it also depends on its behavior for distributions that are moderately different, which is particularly relevant in applications such as particle flows or Approximate Bayesian Computation (ABC). We also refer to our final point, where we highlight the advantages of $d_{KT}$ over MMD in such settings. As a direction for future work, it would be valuable to conduct a theoretical and empirical comparison of the testing power of both metrics under common thresholds. However, we believe these questions involves substantially more work that we leave for a future study.
>
>
> 5. The paper by Mroueh et al., cited in the introduction, hinted at the potential of using this class of IPMs for generative adversarial networks. We acknowledge that it would be interesting to investigate our closed-form expression of the $d_{KT}$ distance leads to an improvement of their numerical results, while they were relying on an approximation of this distance. This may reveal further benefits of $d_{KT}$ in a high-dimensional setting.  Our work serves as a first step in this direction, providing both theoretical foundations and empirical evidence to support the use of this distance.
>
> 6. Let us clarify the experimental setup and the results obtained for ABC. The Kernel Trace distance consistently achieves the lowest mean squared error (MSE), particularly when using a well-chosen value of $\varepsilon$. As we discussed, the fact that MMD accepts samples for very small $\varepsilon$ values—even when other distances reject them—is not necessarily an advantage. In such cases, acceptance under MMD may occur despite model misspecification, whereas rejection by $d_{KT}$ can serve as a useful indicator of such misspecification. We hope the following point will further address any concerns regarding the choice of $\varepsilon$.
>
>
>
> To further demonstrate the advantage of using $d_{KT}$ over MMD in robust learning, we include an additional ABC experiment using linear regression on real data.
>
> We consider normalized data $ (X, Y) = \{(x_j, y_j)\}_{j=1}^n $, and aim to learn a parameter \( \theta \) that minimizes the average squared bias, $ (\theta^\top x_j - y_j)^2 $. While ridge regression is a standard method for this task, we introduce an additional challenge by contaminating 10\% of the training labels—shifting them by +10—to simulate label noise.
>
> As in previous experiments, we apply ABC to sample $ \theta_i \sim \mathcal{N}(0, 0.1 I_d) $, using a zero-centered prior with low variance, consistent with the normalization of the data. We split the data into 90\% training and 10\% testing, and use a Gaussian kernel with bandwidth equal to the data dimension $d$. For each of the 10 000 sampled $\theta_i$, we generate pseudo-labels $\tilde{Y} = \theta_i^\top X + Z $, where $Z \sim \mathcal{N}(0,1)$, i.e., $ P_{\tilde{Y}|X} = \mathcal{N}(\theta_i^\top X, 1) $.
>
> We then rank the sampled $\theta_i$ based on the distance between the generated distribution $(X, \tilde{Y}) $ and the real data $ (X, Y) $, using either MMD or $d_{KT} $. Instead of applying a hard threshold $\varepsilon$, we select the top 10\% (i.e., the 1 000 smallest-distance samples) to avoid sensitivity to threshold choice—see reference [1] and Algorithm 3 in [2] for the quantile-based selection approach.
>
> Below, we report the empirical average squared bias on the test data for the selected $ \theta_i $, as well as for ridge regression. Since our prior has covariance $0.1 I_d $, the corresponding ridge penalty is $ \lambda = 1 / 0.1 = 10 $, though we also include other values of $\lambda $ for completeness.
>
> As shown in the table, except for the first dataset, the Kernel Trace distance consistently outperforms MMD.
>
> | Dataset |        MMD        |        $d_{KT}$        | Ridge $\lambda=1$ | Ridge $\lambda=10$ | Ridge $\lambda=100$ |
> | --- | --- | --- | --- | --- | --- |
> | fertility | 0.69 | 0.70 | 1.40 | 1.16 | 0.75 |
> | servo | 0.92 | 0.87 | 1.39 |1.10 | 0.61 |
> | autompg | 0.59 | 0.44 | 0.32 | 0.30 | 0.30 |
> | yacht | 1.33 | 1.18 | 1.54 | 2.17 | 2.32 |
> | machine | 1.03 | 0.51 | 0.64 | 0.60 | 0.45 |
> | concreteslump | 1.16 | 1.02 | 3.02 | 2.50 | 1.53 |
> | challenger | 0.90 | 0.86 | 3.45 | 3.68 | 1.75 |
> | forest | 1.06 | 0.96 | 0.97 | 0.93 | 0.92 |
> | stock | 1.41 | 1.27 | 2.01 | 1.89 | 1.36 |
> | autos | 1.24 | 0.95 | 0.49 | 0.43 | 0.38 |
> | breastcancer | 1.23 | 1.12 | 0.98 | 1.00 |  1.00 |
> | housing | 0.77 | 0.67 | 0.47 | 0.46 | 0.56 |
> | pendulum | 0.83 | 0.77 | 1.54 | 1.14 | 0.73 |
>
> [1] Biau, G., Cérou, F., \& Guyader, A. (2015). New insights into approximate Bayesian computation. In Annales de l'IHP Probabilités et statistiques
> [2] Robert, C. P. (2016). Approximate Bayesian computation: a survey on recent results
>
>
> We hope our responses have addressed the reviewer’s concerns and that they will consider revising their score.

---

> > ### Comment · Reviewer_S4kf · 2025-08-04
> >
> > I sincerely thank the authors for detailed responses, which address some of my initial concerns. I have also read the comments from other reviewers and acknowledge the theoretical contributions of the paper. I have decided to revise my score to reflect this. However, I remain concerned that the experimental results are not fully convincing, despite the explanations regarding my previous questions. I explain my concerns in detail as follows.
> >
> > 1. **Experimental results in Table 1:**
> >    Table 1 compares MMD and the proposed $d_{KT}$ under different values of $\epsilon$. At $\epsilon = 0.05$, $d_{KT}$ rejects all samples and ABC fails, and MMD accepts 1,092 samples with an ABC MSE of 0.19. At $\epsilon = 0.25$, $d_{KT}$ achieves an MSE of 0.03 and accepts 58 samples, and MMD accepts more samples (2,964) with an MSE of 1.29. At $\epsilon = 0.5$, MMD accepts 6,168 samples with an MSE of 7.47, and $d_{KT}$ accepts 828 samples with an MSE of 0.12. We can observe that both MMD and $d_{KT}$ accept fewer samples as $\epsilon$ decreases. Therefore, for small $\epsilon < 0.05$, MMD may still perform well and achieve a very low MSE (e.g., 0.03 or <0.03) when accepting few samples (e.g., 58 samples or fewer), while $d_{KT}$ fails, as shown in the case where $d_{KT}$ rejects all samples and ABC fails at $\epsilon = 0.05$. In such cases, MMD can perform over a larger range of $\epsilon$ (including $\epsilon < 0.25$), while $d_{KT}$ performs well only within a narrower range. Thus, MMD is more robust in such cases. Furthermore, to achieve a small MSE, both MMD and $d_{KT}$ require careful tuning of $\epsilon$.
> >
> > 2. **Limited comparison across relevant settings:**
> >    The authors do not compare MMD and $d_{KT}$ in the two-sample testing setting, where MMD was originally proposed. Similarly, while the KT metric was first introduced by Mroueh et al. [2017] for evaluating generative models, the authors do not evaluate $d_{KT}$ against the original KT metric in that context either. Instead, they focus on the ABC setting and highlight the advantages of $d_{KT}$ over MMD in this specific application (as noted, "particularly relevant in applications such as particle flows ABC"). I think this comparison may not be entirely fair, and a comparison between $d_{KT}$ and distance metrics specifically designed for ABC would be more appropriate.

---

> > > ### Author Response · Authors · 2025-08-07
> > >
> > > Thank you for getting back to us with comments and having revised your recommendation.
> > >
> > > 1.
> > > - We agree on what you said, that the choice of $\varepsilon$ requires careful tuning. There exists a $\varepsilon$ that narrows down to a few accepted samples. For instance, for the Wasserstein distance it would be very high, while for MMD it would be very low. Our point was to say that not only the produced solution has to be optimal with respect to MSE, but the user should also know if the acquired model actually corresponds to the data before trusting such model. By rejecting all samples for $\varepsilon$ of the order of level of contamination, the Kernel Trace Distance tells the user that either the data has been contaminated (with such level of contamination), or the original family of models considered could not fit the data initially. For the Wasserstein distance, the fine-tuned $\varepsilon$ can be arbitrarly high for different reasons so it is too hard to fine-tune. For MMD, such $\varepsilon$ can be very low, that is why it is very hard to relate it to the order of contamination.
> > >
> > > - To answer your initial concerns and **prevent such need for careful tuning of $\varepsilon$, in our additional real-data experiment we used a quantile threshold**. That means that for both MMD and $d_{KT}$, the top 1000 samples are retained (out of 10 000 for each), the same number for each distance, therefore if MMD was more robust than $d_{KT}$, then its top 10\% models retained should have been very close to the data despite the contamination, but our simulations shows that in most cases $d_{KT}$ has a smaller test error than MMD. We will include those experiments of course in the updated version of the paper.
> > > 2.
> > >
> > > - We carried out a quick test simulation where the 1-Schatten norm ($d_{KT}$) seems to perform  better than the 2-Schatten norm (MMD). We consider two datasets, we assume the first comes from a multivariate Normal and want to test if the second dataset comes from the same distribution as the first.
> > > For 10 000 runs, we have drawn two datasets of size 100 each coming from some (bidimensional) Gaussian distributions. In one case, the two datasets comes from the same distribution: the standard multivariate Normal, which we use to empirically get some threshold $t$ of the 0.95 quantile of the distribution of the distance between the two datasets. In a second case, the first dataset still comes from the Normal centered in zero but the second is drawn from a Gaussian distribution with shifted mean $[0.5,0.5]$. Then we find that the (empirical) probability that such distance is below the threshold $t$, in other words, **the type II error, is $0.054$ for $d_{KT}$ while it is $0.154$ for MMD** (we use the Gaussian kernel with $\sigma=1$).
> > >
> > > However we believe it requires further extensive study with different scenarios. Concerning GAN, the paper by Mroueh et al. *is not the original metric, but a different metric, since it is not the same features* (whether approximated or not) and we also think it requires careful tuning of architecture and training, left for future works.
> > >
> > > - You wrote that "comparison between
> > >  and distance metrics specifically designed for ABC would be more appropriate". However MMD is one of the most studied distance in distance-based ABC since there are several related articles such as [1,2] (also with a slight variation [3,4,5]) as well as one on the energy distance [6] which is a particular case of MMD. The other most known alternative are Wasserstein distance (see below) and the Kullback-Leibler divergence, which is hard to estimate.
> > >  Still, **we added to our table the Wasserstein distance, which does not perform as good as our distance**.
> > >
> > >  | Dataset | MMD | $d_{KT}$ | Wasserstein |
> > > | --- | --- | --- | --- |
> > > | fertility | 0.69 | 0.70 | 0.72 |
> > > | servo | 0.92 | 0.87 | 0.93 |
> > > | autompg | 0.59 | 0.44 | 0.52 |
> > > | yacht | 1.33 | 1.18 | 1.27 |
> > > | machine | 1.03 | 0.51 | 0.61 |
> > > | concreteslump | 1.16 | 1.02 | 1.16 |
> > > | challenger | 0.90 | 0.86 | 0.89 |
> > > | forest | 1.06 | 0.96 | 1.00 |
> > > | stock | 1.41 | 1.27 | 1.34 |
> > > | autos | 1.24 | 0.95 | 0.61 |
> > > | breastcancer | 1.23 | 1.12 | 1.05 |
> > > | housing | 0.77 | 0.67 | 0.72 |
> > > | pendulum | 0.83 | 0.77 | 0.80 |
> > >
> > > We hope it helped clarifying some of your concerns, we thank you again for your time dedicated to the review process.
> > >
> > >  [1] Legramanti, et al. (2025). Concentration of discrepancy-based approximate Bayesian computation via Rademacher complexity.
> > >
> > >  [2] Angelopoulos et al.(2024). Approximate Bayesian Computation with Statistical Distances for Model Selection
> > >
> > >  [3] Park et al. (2016). K2-ABC: Approximate Bayesian computation with kernel embeddings.
> > >
> > >  [4] Chérief-Abdellatif, et al. (2020). MMD-Bayes: Robust Bayesian estimation via maximum mean discrepancy.
> > >
> > >  [5] Dellaporta, et al. (2022). Robust Bayesian inference for simulator-based models via the MMD posterior bootstrap.
> > >
> > >  [6] Nguyen, et al. (2020). Approximate Bayesian computation via the energy statistic.

---

### Official Review · Reviewer_vtMw · 2025-07-02

**Clarity:** 3
**Significance:** 2
**Originality:** 3
**Rating:** 5
**Confidence:** 4

**Summary:**

This paper defines a new kernel trace distance $d_{KT}$ between probability measures. The idea is to embed each distribution as its expected second moment matrix/operator after applying the kernel embedding map, then taking the Schatten 1-norm between them. They show how to compute this explicitly using SVD and the kernel trick. Under some assumptions on the kernel, they show that it is IPM over a class of bounded function with some Lipschitzness-like structure, allowing them to upper bound $d_{KT}$ by TV and some Wasserstein distances. Compared to MMD with the squared kernel (corresponding to the Schatten 2-norm above), they show that it has some nicer normalization properties but still exhibits fast empirical convergence under some standard decay assumptions. Their results are verified with some small experiments.

**Questions:**

1. Since there are so many choices for statistical distances, I personally favor those with some operational meaning or relation to downstream tasks of external interest. E.g. most f divergences tell you something about rates of fundamental testing questions, W1 tells you about estimating Lipschitz test functions and W2 gives rise to rich metric geometry. I don't think this is strictly necessary for acceptance (but it would definitely push me over the bar) - can you think of any way of motivating $d_{KT}$ from first principles? The analogies from quantum are nice but pretty hand-wavy.
2. For transparency, are there any settings where you think $d_{KT}$ should *not* be used over MMD?
3. When working in $\mathbb{R}^d$ with Euclidean structure, we usually center covariance matrices by the mean. You could analogously consider $\tilde{\Sigma}\_\mu = \mathbb{E}\_\mu[(\varphi(x) - \mathbb{E}\_\mu[\varphi(x)])(\varphi(x) - \mathbb{E}\_\mu[\varphi(x)])^\top] = \Sigma\_\mu - \mathbb{E}\_\mu[\varphi(x)] \mathbb{E}\_\mu[\varphi(x)]^\top$ for your setting. Is this a reasonable alternative, and how might it compare to your choice?
4. Do you believe that the rates from Cor 4.3 are tight up to log factors? I would guess so for the second rate at least. I don't think lower bounds are a top priority but they would of course be welcome.

**Ethical Concerns:**

["NO or VERY MINOR ethics concerns only"]

**Final Justification:**

The cubic complexity is indeed a drawback, and the experiments serve more as basic validation of theory than a reason for acceptance on their own right. It is fair to say that significant follow-up work is needed to determine the practical value of this distance. However, I personally found the conceptual contributions nice enough that I lean towards acceptance. My reviewing load featured several works pertaining to new variants of statistical distances - of those, I found this paper to have the most creative and interesting ideas.

**Limitations:**

See second question.

**Quality:**

3

**Strengths And Weaknesses:**

Strengths:
- This is an interesting definition explored through a few different lenses, with advantages over Wasserstein and MMD distances in certain regimes. I occasionally have qualms with this "style" of paper, introducing a new statistical distance, and while I still have some issues, I think this paper is better than the average one of this flavor, through pretty careful comparisons with existing distances.
- The paper was nice to read, I followed all of their arguments on my first read.
- The convergence rates are perhaps the most technical piece of the paper but are well-explained and appear correct to me. Dimension-free rates are always nice.

Weaknesses:
- The question asked on Page 1 --- "How large such a function space could be before the curse of dimensionality kicks in?" --- is rather ill-posed. There are function classes $\mathcal{F}$ which exhibit empirical convergence rates $\mathbb{E}[\\|\\hat{\\mu}\_n - \\mu\\|\_{\\mathcal{F}}] \\lesssim n^{-1/k}$ for every $k$ between 2 and $d$. For example, one can take $\mathcal{F} = \mathcal{F}_k$ to contain all $f:\mathbb{R}^d \to \mathbb{R}^d$ of the form $f = g \circ U$, where $U$ is an orthogonal projection onto a $k$ dimensional subspace and $g$ is 1-Lipschitz. The resulting IPM is the k-dimensional max-sliced (or projection-robust) 1-Wasserstein distance, which exhibits empirical convergence at rate $n^{-1/k}$ when $\mu$ has bounded second moments.
- In general, comparing statistical distances is always a bit subtle, and it is tricky to say when one is better than another. E.g. robustness is a useful property when data is contaminated, but it may lead to overlooking differences when data is clean. Fast statistical rates lead to fewer samples needed to estimate to a given precision, but guarantees at a given precision are often incomparable between different distances. I have some questions below which would help convince me further of the kernel trace distance's utility.
- The computational complexity is okay, but not great. Worst-case, between empirical measures on $n$ points it looks like $O(n^3)$ instead of $O(n^2)$ for MMD.

Minor nits:
- I believe Prop 3.6 has a simpler proof (not that it is complicated) without factor of 2, since $|d_{KT}(P_\varepsilon,Q) - d_{KT}(P,Q)| \leq d_{KT}(P_\varepsilon, P) \leq TV(P_\varepsilon,P) \leq \varepsilon$
- Line 87, "explicit" usually not a verb
- In Assumption 0, I assume "completely" should be "complete", unless "completely separable" is its own property
- Line 156, "note" confused me, used for definition rather than observation as standard. Same thing on Line 266
- For Assumption 2, I was not familiar with "simple" meaning multiplicity one. Apparently somewhat standard, so that's fine. However, if so, the inequalities which follow should be strict.

---

> ### Author Rebuttal · Authors · 2025-07-31
>
> We thank the reviewer for his careful review and positive comments about our paper. We address his main concerns and answer questions below.
>
> ** Weaknesses **
>
> - Thank you for your insightful comment regarding how to achieve different empirical convergence rates through projection-based function classes. We agree that such constructions—like the k-dimensional max-sliced 1-Wasserstein distance—can circumvent the curse of dimensionality to some extent by leveraging low-dimensional structure. We will incorporate this perspective into the revised manuscript.
>
> That said, our focus in posing the original question was more on “natural” (non-compositional) function classes, such as those used in MMD and Kernel Trace Distance, which do not rely on explicit projections onto subspaces. These classes are typically characterized by covering or entropic numbers, and their complexity is measured directly in the ambient space. In this sense, our contribution offers a perspective that is somewhat orthogonal to the one you raised: the convergence behavior of our kernel-based distance may depend not only on the function class but also on properties of the distribution itself (e.g., the decay rate of eigenvalues), rather than solely on dimensionality or projection structure. We have rewritten this part in the introduction for better clarity.
>
> - computational complexity: we acknowledge the greater computational complexity of our distance in comparison with MMD and believe reducing its computational cost is an interesting research direction for further work. Note yet that Sinkhorn's divergence scales as $\mathcal{O}(n^2\epsilon)$ where $n$ is the number of samples and $\epsilon$ is the entropic regularisation parameter, while recently introduced alternative kernel-based divergences to MMD also suffer from this $\mathcal{O}(n^3)$ (see Chazal et al 2024, which was presented last Neurips).
>
> - Thank you for the improvement of factor 2 of Prop. 3.6.
>
> - Thank you also for the spotting the typos. A completely separable space, also known as *second-countable* means that the topology has a countable basis.
>
> ** Questions **
>
> - Q1: A way of motivating $d_{KT}$ from first principles is (see also answer to reviewer tydU) that it is part of a family of distances - IPM using $p$-Schatten norm- which includes the well established MMD, and it is the most "discriminative" inside this family.
>
> - Q2: MMD is a suitable choice over $d_{KT}$  when it sufficiently discriminates between the given distributions, and when computational efficiency is a priority —such as when computing such distances must be run repeatedly or when working with a very large number of samples. As we wrote in the conclusion, one could investigate the use of the Nyström method (subsampling a lesser number of samples) to diminish the computation time of $d_{KT}$.
>
> - Q3: Thank you for the suggestion—considering a centered version of the covariance is indeed an interesting direction. There are two natural approaches: the first, as in your equation, subtracts a constant term from each entry; the second subtracts the outer product of the mean (i.e., centering by removing the mean tensor from the covariance). Thanks to our new difference-kernel trick, the latter is now computationally feasible.
>
> Implementing this second approach involves augmenting the difference kernel matrix by two additional rows and columns (corresponding to the means of each distribution). These new rows and columns are derived from averages of existing rows/columns, so the overall rank remains unchanged, simply adding two zero eigenvalues. Since the extended matrix is constructed from the original difference kernel, we expect its influence on the result to be limited—but it would be worthwhile to verify this empirically. We will add a discussion paragraph for future work mentioning this consideration.
>
> - Q4: The paper [1] establishes lower bounds of order \( O(n^{-1/2}) \) for MMD when using radial kernels. By the inequality \( \|\cdot\|_2 \leq \|\cdot\|_1 \), this lower bound should, \emph{a fortiori}, extend to the kernel trace distance. Moreover, it aligns with the fast convergence rates observed under exponential spectral decay, up to logarithmic factors. For other types of kernels, to the best of our knowledge, corresponding lower bounds for MMD have not yet been established.
>
>
> [1] Tolstikhin, I. O., Sriperumbudur, B. K., \& Schölkopf, B. (2016). Minimax estimation of maximum mean discrepancy with radial kernels. Neurips.

---

> > ### Comment · Reviewer_vtMw · 2025-08-04
> >
> > Thanks for your thorough response.
> >
> > I am leaning towards acceptance but need to take a look at concerns raised by other reviewers first.
> >
> > I'm not completely sure I buy this motivation of "naturality" in your first response, but this is a mostly subjective point aside from the real contributions of the paper (and my score).
> >
> > Regarding centering, do you have any insight into when this would be appropriate? Are there any obvious pros/cons compared to the uncentered approach?

---

> > > ### Author Response · Authors · 2025-08-04
> > > **Concerning centering of the covariance**
> > >
> > > Regarding centering, our current uncentered approach focuses on the full second-moment operator (i.e., the kernel covariance without subtracting the mean), which captures both variance and location-related structure in the RKHS. This makes it more sensitive to differences in location, which can be beneficial in distinguishing distributions that differ not just in spread but in support.
> > >
> > > Centering the covariance operator — by subtracting the mean tensor — would remove this sensitivity to mean location, effectively focusing the distance more on "shape" differences between distributions. This might be preferable in scenarios where differences in location are irrelevant or already handled elsewhere (e.g., in translation-invariant settings).
> > >
> > > A downside is that centering reduces discriminability between certain distributions, especially when location carries meaningful signal. On the other hand, it may improve robustness in cases where location shifts are considered noise.
> > >
> > > In particular, after further experiments, it seems that the spectrum of the centered difference kernel matrix follows the one of the uncentered version, *except for the top highest eigenvalues in absolute value* (concretely, the top positive eigenvalue, and the lowest negative eigenvalue in our experiments). Those values are the predominant ones which makes the distinction between the different distributions.
> > >
> > > For intuition, consider a distribution with a small support: its samples will be mapped to points in the RKHS that are close to each other and to their mean embedding. This means that subtracting the mean from each feature vector removes the global location information and retains only local variations. Now, if we compare two such distributions whose supports are far apart, their mean embeddings in the RKHS will be significantly different—leading to a large distance between them. However, their centered (mean-subtracted) feature representations may appear similar by chance, since these capture only local structure and not the global separation between the distributions.
> > >
> > > We implemented the centered version of $d_{KT}$ with both the Gaussian and Laplacian kernels on 2D Gaussians with different locations on the particle flow experiment. **We empirically verified that the distance between the two different locations does not impact the centered version, which therefore cannot distinguish the distributions**. We have also tried to **reproduce the particle flows with the centered versions of the distance, and as predicted, it does not work at all: particles are unable to move from the source to the target**.
> > >
> > > *Therefore, we thank you for raising this point, and we will definitely comment on this in our paper.*

---

> > > > ### Comment · Reviewer_vtMw · 2025-08-04
> > > >
> > > > Thank you for looking into this, these are nice observations which would be welcome in the revision.
> > > >
> > > > I will need to take a closer look at the discussion with other reviewers, but you have addressed all of my immediate concerns and I am leaning towards acceptance.

---

> > > > > ### Author Response · Authors · 2025-08-05
> > > > > **Thank you**
> > > > >
> > > > > Thank you for your enriching comments and the whole extensive discussion.

---

### Official Review · Reviewer_tydU · 2025-07-03

**Clarity:** 3
**Significance:** 4
**Originality:** 4
**Rating:** 5
**Confidence:** 3

**Summary:**

This paper proposes a new statistical divergence inspired by "quantum statistics".
For a given "feature map" $\varphi(x)$, the new divergence "kernel trace (KT) distance" between two measures $\mu$ and $\nu$ is defined as the Shatten 1-norm of the difference between the covariances of the feature map with respect to the two measures:
$\|\|\Sigma\_\mu-\Sigma\_\nu\|\|\_1$, where $\Sigma\_\mu\triangleq \mathbb{E}_{x\sim \mu} [\varphi(x)\varphi(x')^*]$.
When the feature map $\varphi(x)$ is associated with a PSD kernel $k(x,x')$, i.e., $k(x,x')=\langle \varphi(x),\varphi(x')\rangle$, then the KT distance can be computed by the kernel trick, similar to the standard kernel methods (Section 2.3).
The relations of the KT distance to IPMs (TV and Wasserstein distances) and other quantum statistical divergences (kernel BW and kernel KL divergences) are also established (Section 3.1). Its robustness is also discussed.
The convergence rate of the empirical measure to the underlying measure in the KT distance is established in Section 4. Under a certain spectrum decay assumption of the target distribution, the parametric rate $O(n^{-1/2})$ is nearly attained. As a byproduct, the result also readily implies the dimension-free rate of the KBW distance via its equivalence to the KT distance.
The authors applied the KT distance for ABC and particle flow experiments.

**Questions:**

- In general, one can consider a family of divergences defined as a Schatten-$p$ norm for any $p\ge 1$, where $p=1$ gives the KT distance and $p=2$ gives the MMD distance with the kernel $k^2$. Is there any reason that $p=1$ is particularly special from this family? For example, I wonder if the robustness result in Proposition 3.6 would hold for any $p\ge 1$. Would the distance with larger $p$ have less discriminative power in general? It'd be very instructive if the authors can discuss this general family for more insights.
- In Proposition 3.5, what is the meaning of the first sentence? Do the authors mean to consider the mixtures for distributions that satisfy orthogonality between covariances?
- It is claimed in the last paragraph that `[The KT distance] is the greatest in a family of kernel-based IPM including MMD`, but how can I see this directly? Maybe I am missing something, but it seems not obvious to me.
- In line 328, is `For the proposed distance $d\_{KT}$` a typo?

## Typos
- Line 104: it ~is~ could also be...
- Line 147, put , after `expressive`
- Line 217, the ``internal energy'' $\|\|\Sigma\_\mu\|\|_2$ ...
- After line 274, $\phi$ should be $\varphi$, which is the feature map of the considered RKHS?
- The subsection Section 4.1 seems to be not necessary as this is the only subsection in Section 4.

## Suggestions
- Reading Eq. (5), I had to try to find the definition of $F(A,B)$ again. It'd be helpful if the definition of fidelity can be recalled in the revision.
- Please consider revising and expanding the section on experiments in the camera-ready, if accepted.


Overall, I think the manuscript has an interesting contribution, and I will be happy to increase the score if the authors promise to revise the manuscript and address the comments.

**Ethical Concerns:**

["NO or VERY MINOR ethics concerns only"]

**Final Justification:**

The authors' rebuttal sufficiently addressed my original concerns regarding clarity. Incorporating these points (and those raised by other reviewers, especially on experiments) will greatly improve the quality of the paper.
I've increased the Clarity score from 2 to 3 given that the authors will revise the manuscript accordingly, and also increased the overall rating from 4 to 5.

**Limitations:**

The limitations of this method are properly mentioned.

**Paper Formatting Concerns:**

The alignment of the caption of Table 1 doesn't seem to comply the suggested format.

**Quality:**

3

**Strengths And Weaknesses:**

## Strengths
The paper is overall well written and easy to understand (except a few parts).
The writing is technically solid and mostly easy to understand.
The idea of considering the Schatten-1 norm of the difference between the covariances sounds natural and elegant.
The proposed distance has a better discriminative power than MMD.
Its relationship to other divergences are well studied.

## Weaknesses
Even though the exposition is overall easy to understand, I believe some parts can be revised to improve readability.
In particular, I cannot understand what the authors wish to convey in Section 3.2.
Also, the writing in `Section 5 Experiments` is much less clear than other parts. For example, the explanation on ABC was not comprehensible for me, who first learned the approach from this paper, based on the reading of this paper. The low score is mostly due to the lack of clarity on these parts.

---

> ### Author Rebuttal · Authors · 2025-07-31
>
> We thank the reviewer for his careful review and positive comments about our paper. We address his main concerns below.
>
> ** Questions **
>
> - The 1-Schatten norm holds a special place among the $p$-Schatten norms for $p \geq 1$, as it is the largest in that family (since $\|\cdot\|_s \leq \|\cdot\|_p$ for $p \leq s$). This has important implications for Integral Probability Metrics (IPMs). Each $p$-Schatten norm admits a dual formulation via Hölder’s inequality, using the conjugate exponent $q$ such that $1/p + 1/q = 1$. In particular, when $p = 1$, the dual norm is the $\infty$-norm, which is the least restrictive and thus defines the largest possible set of discriminator functions in the IPM. In this sense, higher values of $p$ lead to weaker discriminative power, since the function class has the same structure as $\mathcal{F}_1$, but with the operator $U$ constrained in the $q$-norm rather than the $\infty$-norm. Furthermore, the 2-Schatten norm corresponds to MMD with kernel $k^2$. But since any positive definite kernel $k$ allows a square root $k'$ such that $k = (k')^2$, we can see MMD as a special case within this family. This leads to the conclusion that the KT distance is, in fact, the **largest** among this class of kernel-based IPMs, including MMD.  The proof of Proposition 3.6 works indeed for all $p\geq 1$ (see l939 in the Appendix) because of the inequality between the norms.
>
> - In Prop. 3.5, indeed, the orthogonality condition is $\Sigma_{\mu_1} \perp \Sigma_{\mu_2}, \Sigma_{\mu_1} \perp \Sigma_{\nu_2}, \Sigma_{\nu_1} \perp \Sigma_{\mu_2}, \Sigma_{\nu_1} \perp \Sigma_{\nu_2}$.
>
>
> - Thank you for noticing the typos, we have corrected them. At line 328, this was indeed a typo — we intended to write “For the traditional likelihood” to indicate that it does not apply in this context. We apologize for the confusion. We hope this will clarify why standard Bayesian method (using Bayes'rule update) fails and robust ABC using our novel distance is necessary. $\phi$ was indeed $\varphi$ and we have removed the unnecessary subsection 4.1. We will also recall the fidelity with the Fuchs-van de Graaf's inequalities.
>
> ** Weaknesses **
>
> Thank you for suggesting writing improvements. Below we clarify the message of Section 3.2 and ABC in Section 5.
>
> - Section 3.2: This section of the paper introduces the concept of normalized energy in the context of comparing probability measures via RKHS-based operators. We highlight a key distinction between our proposed Kernel Trace (KT) distance and the widely used Maximum Mean Discrepancy (MMD), that both a covariance mean embedding. The information about the distribution is contained in the eigenvalues (similar to probability mass) and eigenvectors (similar to distribution support) of the RKHS density operator. The MMD relies on the Schatten 2-norm and is sensitive to the internal "energy"
> $$||\Sigma_\mu|| = \int\int k(x,y)^2 \mu(x)  \mu(y)dx dy = \int_\mathcal{X} \int_\mathcal{Y} e^{-||x-y||^2/\sigma^2} \mu(x)  \mu(y)dx dy$$, (where $||\cdot||$ denotes the 2-norm here), which is itself related to the variance (or "spread") of distributions. It depends not only on the distribution but also on the kernel's bandwidth. Since our focus is on the distance between two embeddings—rather than representing a single embedding—we showed in Section 3.2 that embeddings with low energy can yield a low MMD value even when they differ significantly. This is because the bandwidth parameter cannot be simultaneously tuned to capture both scale and difference effectively.
> In contrast, the $d_{KT}$ distance uses the Schatten 1-norm, which equals one for all distribution (for all $\mu$, $|\Sigma_{\mu}| = 1$ where $|\cdot|$ denotes the 1-norm here) since we assumed the trace of RKHS density operators is 1. This normalization ensures that the KT distance focuses purely on the separation between distributions, not on their individual energy levels. Our empirical results (e.g., Figure 1) show that KT distance behaves better in this sense than MMD when comparing two fixed distributions while increasing the kernel bandwidths:
> $d_{KT}$ decreases as expected, while MMD is not monotone. In Figure 2, one can see that the $d_{KT}$ distance attains its theoretical maximal value (2), while MMD achieves a much smaller value (value 2 would be approachable by two far-away Dirac). In Figure 3, our distance drops below its maximum value of 2 only when the supports begin to significantly overlap (i.e. when the std deviation $s$ is of the order of the distance between the mean); in contrast, MMD decreases much earlier, as it naturally diminishes with increasing variance. Furthermore, as a second step, we demonstrate (Proposition 3.5) that for mixtures of distributions, the $d_{\mathrm{KT}}$ distance scales linearly with the component-wise distances, thereby preserving discriminability, whereas MMD tends to lose resolution (by a factor 2) due to the averaging effect inherent in its energy-based formulation. Generally, this section is intended to justify why $d_{KT}$ leads to a more stable and robust behavior, making it more reliable in scenarios where MMD’s performance deteriorates. We will highlight those clarifications in paper.
>
>
> -ABC in Section 5: Traditional Bayesian methods rely on computing the posterior distribution over models by updating a prior based on observed data. This can be computationally expensive, and a misspecified prior may lead to incorrect inference. As an alternative to Bayes' rule in such cases, one can use Approximate Bayesian Computation (ABC).
>
> A key reference is Algorithm 1 in Appendix B.2. ABC is a Monte Carlo method in which models $p_{\theta_i}$ are sampled from the prior distribution, repeated for $i = 1, \dots, T$ iterations. Each sampled model is then evaluated for its fit to the observed data using a criterion such as a distance metric. The subset of models with the best evaluations is retained, forming an approximate posterior distribution over models.
>
> In our setting, rather than using a likelihood, we compare distributions using a distance metric, and the threshold $\varepsilon$ controls the degree of model mismatch tolerated. This is particularly useful in scenarios with contaminated data, even when the model is otherwise correct. Because our distance metric tolerates a fraction $\varepsilon$ of mass to be mismatched (rather than, for instance, tolerating a shift in the mean), it offers greater robustness and efficiency—especially since the mean is not a robust statistic.
>
> We are also expanding this section with additional ABC experiments; see our response to Reviewer S4kf at the end of the rebuttal for details.
>
>
> We hope our responses have addressed the reviewer’s concerns and that they will consider revising their score.

---

> > ### Comment · Reviewer_tydU · 2025-08-05
> >
> > I appreciate the authors' thoughtful rebuttal. Please carefully address all these points in the manuscript, which I believe will help highlight the contribution of the paper. Expanding the experimental setup with details will help readers with little exposure understand the benefit of the approach.
> > Since the authors have sufficiently addressed my questions, I've increased my overall rating to acceptance given that the manuscript will be properly updated.

---

> > > ### Author Response · Authors · 2025-08-06
> > > **Many thanks**
> > >
> > > We thank you again for your remarks that helped us improve the readability of our article and for increasing your appreciation in consequence. We are adding all of those clarifications. Following the advice of reviewer B5gM, we will also split section 3.2 in two parts for better clarity, and while adding our new experiments, we will insist more on the basics of ABC.

---

### Decision · Program_Chairs · 2025-09-17

**Decision:**

Reject

**Comment:**

This paper defines a new distance, denoted as $d_{KT}$, between probability distributions via the Schatten 1-norm of their kernel covariance operators, with improved discriminating power and robustness to hyperparameters. The distance is validated on small-scale
approximate Bayesian computation and particle flow simulations, with limited comparisons to MMD and Wasserstein distances.

The reviewers acknowledge the novel definition as conceptually creative and well ground in theory. The reviewers recognize the rigor and clarity of the theoretical contributions, noting that the authors have proved the fundamental properties of $d_{KT}$ including its discriminative power, robustness, and convergence.

Although the reviewers assigned high scores, their reviews consistently reflected important weaknesses. Several reviewers (S4kf, MQLz,B5gM, and in the final justification, vtMw) highlighted that the empirical validation is insufficient to confirm the claimed advantages of $d_{KT}$ and raised concerns about the cubic computational complexity (vtMw, B5gM). This is a critical criterion for a distance intended for practical use. I encourage the authors to address this feedback and redesign the experiments to convincingly demonstrate the robustness, discriminative power, and to validate that the computational cost is acceptable in scalable settings.

The general methodology of using covariance operators to define distances has been studied in prior work. While the proposed distance is interesting and considered novel by the reviewers, its positioning in this line of research should be clarified and strengthened. In particular, a more direct comparison with closely related distances (e.g.  Mroueh et al. [2017]; Kernel Methods on Covariance Operators, Minh and Murino, 2018; Maximum Mean and Covariance Discrepancy for Unsupervised Domain Adaptation, Zhang et al. 2019) would help to better establish the contribution of the current work. The latter two references are suggested by the AC as examples of related definitions.

The reviewers have discussed the pros and cons of the contributions thoroughly with the AC. The authors' rebuttal and related discussions have been considered carefully. In light of the above weaknesses and the high competition in related areas, I recommend rejection.